# The life cycle of submesoscale eddies generated by topographic interactions

Mathieu Morvan[1], Pierre L'Hégaret[1], Xavier Carton[1], Jonathan Gula[1], Clément Vic[1], Charly de Marez[1], Mikhail Sokolovskiy[2], and Konstantin Koshel[3]

[1]LOPS, Univ. Brest-CNRS-IFREMER-IRD, IUEM
[2]Institute of Water Problems of the RAS, Ul Gubkina 3, Moscow, 199333, Russia, Shirshov Institute of Oceanology of RAS, 36 Nahimovskiy pr., Moscow, 117997, Russia
[3]V.I.Il'ichev Pacific Oceanological Institute, 43, Baltiyskaya Street, Vladivostok, 690041, Russia

**Correspondence:** Mathieu Morvan (mmorvan3@univ-brest.fr)

**Abstract.**

The Persian Gulf Water and Red Sea Water are salty and dense waters flowing at intermediate depths in the Gulf of Oman and the Gulf of Aden respectively. Their spreading pathways are influence by mesoscale eddies that dominate the surface flow in both semi-enclosed basins. *In situ* measurements combined with altimetry indicate that Persian Gulf Water is stirred in the form of filaments and submesoscale structures, by mesoscale eddies. In this paper, we study the formation and the life cycle of intense submesoscale vortices and their potential impact on the spreading of Persian Gulf Water and Red Sea Water. We use a primitive-equation three-dimensional hydrostatic model at a submesoscale-resolving resolution to study the evolution of submesoscale vortices. Our configuration idealistically mimics the dynamics in the Gulf of Oman and the Gulf of Aden: a zonal row of mesoscale vortices interacting with north and south topographic slopes. Intense submesoscale vortices are generated in the simulations along the continental slopes due to two different mechanisms. First, intense vorticity filaments are generated over the continental slope due to frictional interactions of the background flow with the sloping topography. These filaments are shed into the ocean interior and undergo horizontal shear instability that lead to the formation of submesoscale coherent vortices. The second mechanism is inviscid and features baroclinic instabilities arising at depth due to the weak stratification. Submesoscale vortices subsequently drift away, merge and form larger vortices. They can also pair with opposite signed vortices and travel across the domain. They eventually dissipate their energy *via* several mechanisms, in particular fusion into the larger eddies or erosion on the topography. Since no submesoscale flow clearly associated with the fragments of Persian Gulf Water was observed *in situ*, we modelled Persian Gulf Water as Lagrangian particles. Particles patches are advected and sheared by vortices and are entrained into filaments. Their size first grows as the square root of time : a signature of the merging processes. Then it increases linearly with time, corresponding to their ballistic advection by submesoscale eddies. On the contrary, without intense submesoscale eddies, particles are mainly advected by mesoscale eddies; this implies a weaker dispersion of particles than in the previous case. This shows the potentially important role of submesoscale eddies in spreading Persian Gulf Water and Red Sea Water.

# 1 Introduction

Mesoscale eddies $[\mathcal{O}(10\text{-}100)\text{km}]$ are ubiquitous in the world's oceans. The mesoscale eddy field dominates the energy content of oceanic currents at sub-inertial frequencies and is crucial to large-scale budgets of heat and geochemical tracers. Although their generation mechanisms are clearly identified and quantified, largely stemming from large-scale current instabilities, how they dissipate their energy remains unclear (Ferrari and Wunsch, 2009). The interaction of these mesoscale vortices can lead to the formation of smaller scale features (Carton, 2001). Submesoscale eddies $[\mathcal{O}(1\text{-}10)\text{km}]$ can be generated as the result of frontal instabilities (Capet et al., 2008a, b, c) feeding off the energy of the mesoscale currents. Recently, numerical models highlighted one possible mechanism for mesoscale energy dissipation, which is the interaction of mesoscale eddies with the underlying seafloor topography (Molemaker et al., 2015; Gula et al., 2015, 2016; Vic et al., 2015). This specific interaction also leads to the formation of submesoscale eddies. In this regime, the influence of stratification is still important, but the Coriolis force is less prevalent in the horizontal momentum budgets (viz. the Rossby number of the flow is not always small; McWilliams (2016)). Submesoscale eddies have been sampled for decades, yet their lifecycle (from generation to dissipation) is poorly understood (McWilliams, 1985; D'Asaro, 1988; Bosse et al., 2015).

The Arabian Sea is home to an energetic mesoscale eddy field (Carton et al., 2012; Vic et al., 2014; L'Hégaret et al., 2015). In particular, they can interact with the salty and dense waters of the outflows from the Persian Gulf and from the Red Sea (Bower et al., 2002; Al Saafani et al., 2007). These salty and dense waters flow out of these seas into the Gulf of Oman and into the Gulf of Aden and settle at 250-300 m and 600-1000 m depths, respectively (Pous et al., 2004a; Bower and Furey, 2012). These two gulfs also receive Rossby and Kelvin waves, and vortices, propagating westwards from the Arabian Sea (L'Hégaret et al., 2013). Mesoscale vortices divert the outflow paths away from the coast, advect them along curved trajectories around the eddy rims, elongate these outflows as salty filaments and finally, can break these filaments into small eddies (Pous et al., 2004b; Carton et al., 2012). The chaotic dispersion of a tracer by a mesoscale vortex row in a realistic numerical simulation of the Japan Sea has previously been studied by Prants et al. (2011). However, the details of the spreading mechanism, involving submesoscale eddies, has not been explored.

In this paper, we focus on the life cycle of the submesoscale eddies generated by mesoscale eddy-topography interactions, combining observational data from dedicated campaigns and idealized numerical simulations. Vic et al. (2015) studied the case of a mesoscale dipole along a continental shelf, reminiscent of the situation described in L'Hégaret et al. (2013). But satellite observations during the spring inter-monsoon and the summer monsoon reveal the presence of vortex rows with alternate polarities in the Gulf of Aden and in the Gulf of Oman. A vortex row is more efficient than a single dipole to form small eddies, but it is also more prone to destroying them by shearing and stretching effects. These mechanisms will be studied here. We further quantify how these submesoscale eddies are instrumental in the spreading of the dense overflow waters modelled here as a passive tracer. We compare and analyze a set of three numerical experiments with different bottom boundary conditions (with or without bottom drag and bottom boundary layer) and with different topographies (with or without cape).

The data and numerical model used in this study are presented in Section 2. Observations of submesoscale eddies and fragments formed from the Persian Gulf Water outflow in the Gulf of Oman in spring 2014 are presented in Section 3.1. Then, generation and life-cycle of the submesoscale eddies due to the interaction of a vortex row with continental slopes is analysed using idealized numerical simulations in sections 3.2, 3.3 and 3.4.

## 2   Material and methods

### 2.1   The Physindien 2011 campaign

The Physindien experiment was carried out in spring 2011 to study the Persian Gulf Water outflow current in the Gulf of Oman, and to monitor the eddies there and south of the Arabian Peninsula. The Beautemps-Beaupre research vessel deployed

CTD/ADCP (Conductivity Temperature and Depth probe/Acoustic Doppler Current Profiler) at different locations, to record vertical profiles of pressure, salinity, temperature and currents; it also towed a Seasoar, a platform carrying two CTD's, undulating between 0 and 350 m depths behind the ship. The measurement accuracies after processing and validation on other (independent) sensors were $10^{-3}\,°\mathrm{C}, 5\ 10^{-3}\,\mathrm{psu}$. The horizontal velocity was obtained with a $38\,\mathrm{kHz}$ ship-mounted ADCP; this device measures currents from the surface to about 1000 m depth; the depth range depends on the matter in suspension

in seawater, which can reflect the acoustic signal. The Long Time Average data are used, giving one profile every 10 minutes. The accuracy of the horizontal components of velocity is $5 \times 10^{-3}\,\mathrm{m\,s^{-1}}$. The characteristics of the Persian Gulf Water are highlighted *via* the vertical sections of temperature, salinity and density.

### 2.2   Numerical set up

To investigate the origin and the dynamics of submesoscale vortices, we set up idealized simulations of a row of four mesoscale

surface intensified vortices with alternate polarities, in a zonally periodic channel, by using the Coastal and Regional Ocean COmmunity (CROCO) model at submesoscale-resolving resolution on a f-plane.

The CROCO model is a primitive equation model solving the hydrostatic Boussinesq Navier Stokes equations with Coriolis acceleration due to the Earth rotation. On the horizontal, the momentum and tracers equations are solved by using the $5^{\mathrm{th}}$-

order upstream-biased advection scheme. On the vertical, the momentum and tracers equations are solved by using the splines vertical advection scheme. The extent of the channel is $[600\,\mathrm{km}; 200\,\mathrm{km}; 2000\,\mathrm{m}]$ in the x-,y- and z-directions respectively. The horizontal grid spacing is $1\,\mathrm{km}$. On the vertical, 100 submesoscale-resolving $\sigma$-levels stretched at the bottom are used (the surface and bottom stretching parameters are respectively $\theta_s = 2$ and $\theta_b = 6$). A special care has to be paid to the grid stiffness in order to limited the hydrostatic pressure gradient errors. We performed sensitivity experiments by varying both the

number of $\sigma$-levels and the bottom stretching coefficient. The hydrostatic pressure gradient errors appear as very intense noise in the vorticity field at the grid-scale in some experiments. The simulations we show in the paper do not exhibit any spurious

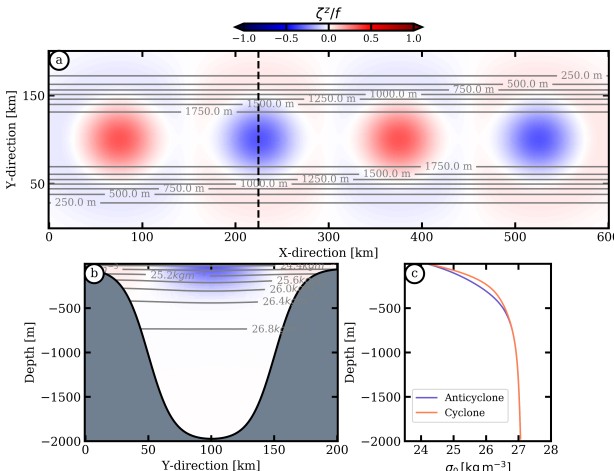

**Figure 1.** Initial state: (a) Vertical relative vorticity (normalized by the Coriolis frequency) for the four mesoscale surface intensified eddies. Black contours are the isobaths. (b) Vertical cross-section of an anticyclone. (c) Vertical profiles of density associated with the cores of a mesoscale cyclone (orange) and of an anticyclone (blue).

numerical noise with the previous parameters.

The mean stratification is chosen such that the deformation radius is about 40 km, as computed by Chelton et al. (1998) in the Gulf of Oman. The mean vertical profile of temperature is

5 $$T(z) = T_0 \exp(z/H), \tag{1}$$

where $T_0 = 28\,^\circ\text{C}$ and $H = 2000\,\text{m}$ (mean depth of the Gulf of Oman). $T_0$ is chosen so that the surface density field is representative of the Gulf of Oman. The salinity is kept to $35\,\text{psu}$ over the water column.

We initialize the flow with four mesoscale vortices, with alternate polarity along the channel axis (Figure 1). The radial and 10 vertical profiles of circular velocity of each vortex are :

$$v_\theta(r,z) = \pm \frac{v_0 r}{R} \exp\left(-\frac{r^2}{R^2}\right) \exp\left(-\frac{z^2}{D^2}\right), \tag{2}$$

with $v_0 = 0.5\,\text{rad}\,\text{s}^{-1}, R = 50\,\text{km}, D = 300\,\text{m}$, typical values found in the Gulf of Oman (de Marez et al., 2019). Then the velocity field of the four mesoscale eddies is balanced with their pressure field using the cyclo-geostrophic equilibrium and following the procedure described in Ciani et al. (2016). The density anomalies associated with the vortices are directly com-15 puted using the hydrostatic equilibrium. The vortices are separated from each other horizontally by $150\,\text{km}$. The distance between the vortex cores and the northern and southern edges is $100\,\text{km}$.

To characterize the impact of the frictional effects at the bottom and of the topography shape on the submesoscale eddy generation, we design three experiments with different bottom boundary conditions (with or without bottom drag and bottom boundary layer) and with different topographies (with or without a cape). In the first experiment (NO-BBL), we use a free slip bottom boundary condition with no bottom boundary layer mixing and parameterize the smooth bathymetry representative of the continental slope in the Gulf of Oman as:

$$h = h_{shelf} + \frac{1}{2}\left(H - h_{shelf}\right)\left(\tanh\left(\frac{y - y^*}{W_y}\right) - \tanh\left(\frac{y - y_N}{W_y}\right)\right),\tag{3}$$

where $h_{shelf} = 100\,\mathrm{m}$, $H = 2000\,\mathrm{m}$, $y^* = y_S = 50\,\mathrm{km}$, $y_N = L_y - y_S$, $L_y = 200\,\mathrm{km}$ and $W_y = 20\,\mathrm{km}$.

In the second experiment (BBL), the same bathymetry is used but we add a bottom drag and a vertical mixing at the bottom with a KPP scheme (Large et al., 1994). The bottom drag is implementend *via* the Von-Karman quadratic bottom stress formulation defined as:

$$\tau_{\mathbf{b}} = \rho_0 C_D \, || \, \mathbf{u} \, || \, \mathbf{u},\tag{4}$$

where $\rho_0$ is a reference density. $C_D$ is the drag coefficient varying as:

$$C_D = \left(\frac{\kappa}{\log\left(\Delta z_b / z_r\right)}\right)^2,\tag{5}$$

where $\kappa$ is the Von-Karman constant equal to 0.41, $z_r$ is the roughness parameter equal to $1\,\mathrm{cm}$ and $\Delta z_b$ is the thickness of the lowest layer of the grid. These parameterizations lead to the formation of a turbulent bottom boundary layer as soon as a current flows over a sloping bottom topography.

Then, in the third experiment (BBL-CAPE), we add a cape in the topography defined by eq. 3 by setting $y^* = y_S + y_0 \exp\left(-\left(\frac{x - x_0}{W_x}\right)^2\right)$, $y_0 = 40\,\mathrm{km}$ and $W_x = 20\,\mathrm{km}$. We use the same parameterizations of the vertical mixing at the bottom and the bottom drag as in the BBL experiment.

Finally, we use the ARIANE tool (Blanke and Raynaud, 1997) to study the impact of the production of submesoscale eddies on the spread of Lagrangian particles from the coast towards the center of the gulf. The Persian Gulf outflow Water is not explicitly included in the model since no submesoscale velocity structure associated with the fragment of Persian Gulf Water was observed in the *in situ* measurements (see section 3.1). Also, we are not interested in the dynamics of the outflow itself for which periodic boundary condition in the x-direction would have been problematic. Rather we focus on the impact of submesoscale processes on the Persian Gulf outflow Water and thus Lagrangian particles are used to model this water mass. ARIANE is a computational tool dedicated to the offline calculation of 3D streamlines in the output velocity field of model whose equations are based on volume conservation. Transports of water masses or currents are deduced from the displacement of numerical Lagrangian particles.

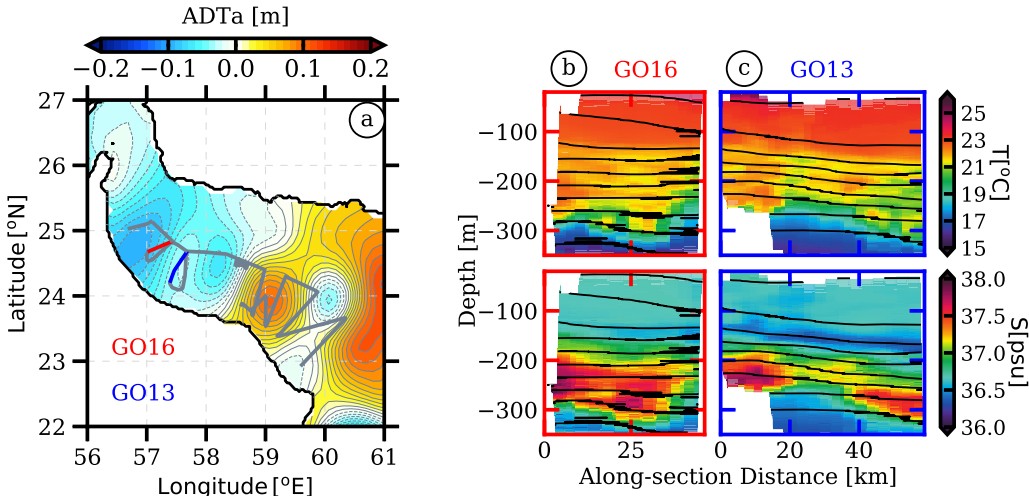

**Figure 2.** a) Map of the Absolute Dynamic Topography anomalies in the Gulf of Oman averaged over the duration of the cruise processed by CLS-Argos on a $1/8^\circ$ Mercator grid. The solid grey lines indicate the locations of the Seasoar sections. The solid orange and blue lines stand for the sections GO16 and GO13 respectively. b-c) Temperature (top) and salinity (bottom) vertical sections of GO16 and GO13 sections respectively showing the warm and salty PGW outflow settled down to about $250\,\mathrm{m}$ depth in the Gulf of Oman. Black contours stand for isopycnals.

## 3 Submesoscale eddy life cycle

### 3.1 Observations in the Physindien 2011 *in situ* data

In spring 2011, a vortex row of alternate polarity was present along the axis of the Gulf of Oman. The Persian Gulf Water (PGW) outflow was located along the continental slope of Oman (south of the Gulf of Oman). The vortex row in the Gulf of Oman shed small fragments of PGW into the basin interior. The Physindien 2011 Seasoar section crossed the outflow and the PGW fragments (see Figure 2a).

In particular sections GO13 and GO16 (see Figure 2b,c), show filaments of warm and salty water extending from the continental slope offshore over $50$ and $60\,\mathrm{km}$ respectively. These filaments have a temperature above $21\,^\circ\mathrm{C}$ and a salinity reaching $37.5\,\mathrm{psu}$, characteristic of the PGW outflow at this location (L'Hégaret et al., 2015). The measurements from the vessel-mounted ADCP (not shown) do not exhibit any velocity structure associated with the filaments of PGW. By considering the horizontal resolution of the vessel-mounted ADCP ($\sim 2\,\mathrm{km}$), this suggests that, at these locations, the PGW can be considered as a passive tracer.

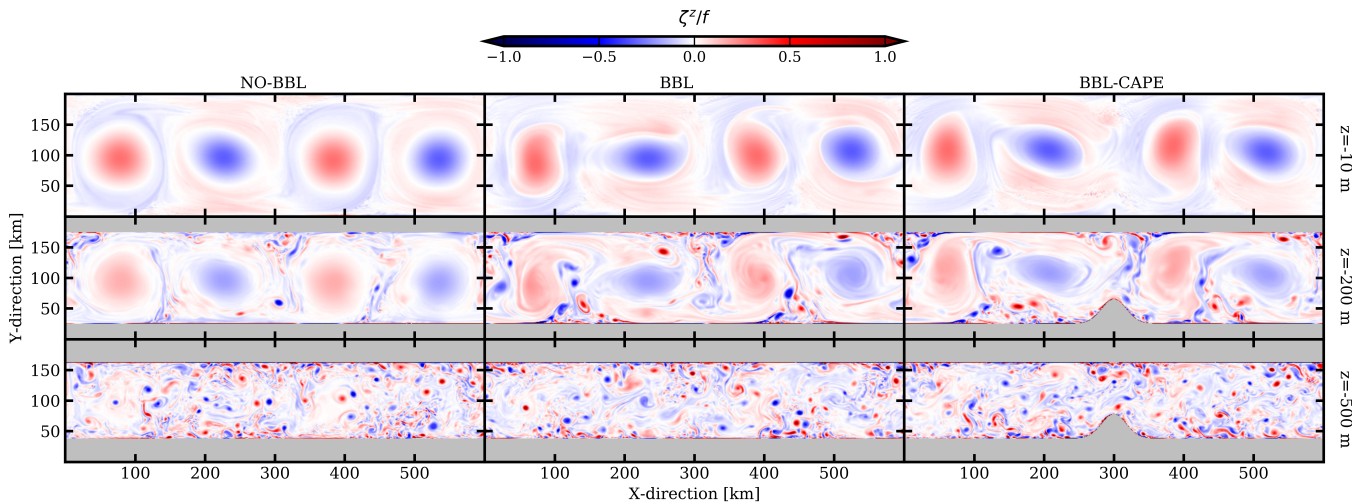

**Figure 3.** Vertical relative vorticity field normalized by the Coriolis frequency at day 200 and 10, 200 and 500 meter depth for the NO-BBL, BBL, and BBL-CAPE experiments.

Both filaments are located between a cyclone (West) and an anticyclone (East). The PGW is advected offshore by the surface intensified mesoscale eddy. In the filaments, the structure is more complex, with an alternation of more or less salty water. This suggests that filaments could break into small vortices as they are advected by surface intensified mesoscale eddies.

### 3.2 Production of submesoscale eddies in the simulations

Numerous submesoscale vortices are generated in the numerical simulations. They are essentially found below $100\,\mathrm{m}$ depth, which corresponds to the depth of the shelf break. Submesoscale vortices originate from the interaction of mesoscale eddies with the sloping topography. In Figure 3, we show the relative vorticity field normalized by the Coriolis frequency at day 200 for the three numerical experiments near the surface, at $200$ and $500\,\mathrm{m}$ depth. The surface flow is dominated by mesoscale eddies for all experiments. At $200\,\mathrm{m}$ depth, submesoscale filaments and vortices are visible around mesoscale eddies. They are more abundant and intense in the experiments in which both the bottom KPP and the bottom drag were active (i.e.: BBL and BBL-CAPE). This indicates that most of these submesoscale vortices are generated by bottom frictional effects. The absolute value of the relative vorticity associated with the submesoscale vortices is close to the planetary vorticity ($|\zeta^z/f| \sim 1$). The flow is dominated by an energetic submesoscale turbulence at $500\,\mathrm{m}$ for all experiments. Submesoscale filaments and vortices are also generated in NO-BBL, where there are no frictional effects at the bottom. This suggests that another mechanism leading to the production of small vortices is at play. This inviscid mechanism is related to baroclinic instabilities. It will be presented in Section 3.3.

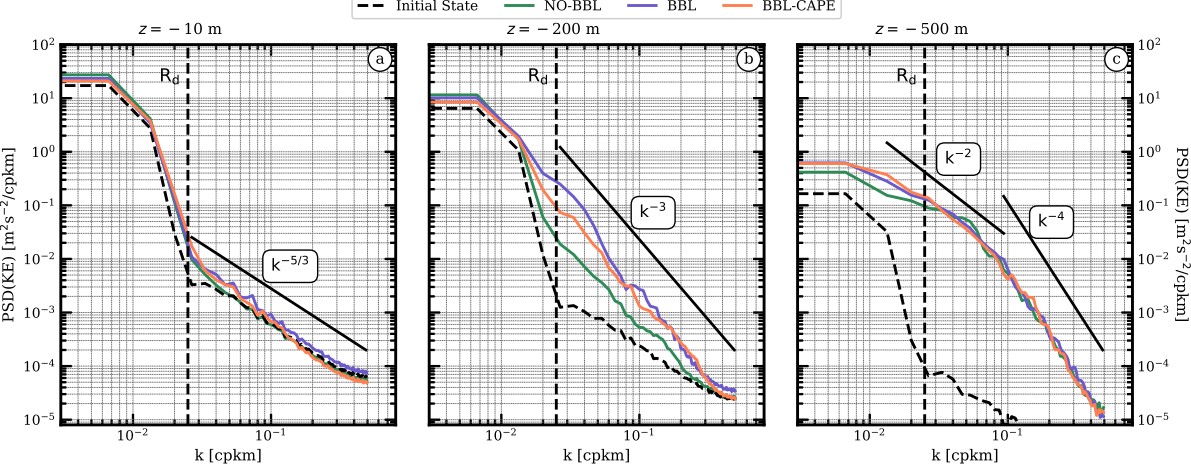

**Figure 4.** Kinetic Energy (KE) spectra comparison of the three experiments: NO-BBL (green), BBL (blue), BBL-CAPE (orange). Spectra are computed at (a) 10, (b) 200 and (c) 500 meter depth at day 200. Black lines stand for benchmarks of power laws.

In Figure 4, we compare the Power Spectrum Density of horizontal velocities for the three experiments (NO-BBL, BBL and BBL-CAPE). Consistently with the previous results, we observe a net increase of the kinetic energy below the Rossby deformation radius ($R_d$) at depth for all simulations. This increase is the signature of subsurface submesoscale eddies. In the two simulations with bottom KPP and bottom drag (i.e.: BBL and BBL-CAPE), and in accordance with Vic et al. (2015), the part of the kinetic energy contained at the submesoscale is larger at 200 m. Near the surface, we observe a slight increase of the kinetic energy below the Rossby deformation radius due to the surface signature of the subsurface submesoscale eddies. Below $500\,\mathrm{m}$ depth, the shapes and the slopes of the spectra are similar for the 3 experiments. The flatter $k^{-2}$ slope at horizontal scales ranging between 100 and $10\,\mathrm{km}$ indicate an active submesoscale turbulence field and a submesoscale source of energy at around $10\,\mathrm{km}$.

## 3.3 Processes of submesoscale eddy generation

### 3.3.1 Baroclinic instability at depth

The first mechanism is an inviscid mechanism visible below $300\,\mathrm{m}$ in all experiments. This mechanism is related to the baroclinic instability. Baroclinic instabilities are triggered by the mesoscale eddy velocity field acting as the perturbation along the sloping bathymetry. This can be seen in the NO-BBL experiment, in the absence of bottom frictional effects. We characterize

this instability by computing the energy transfer terms (Gula et al., 2016), i.e.: the Horizontal Reynolds Stress (HRS), the Vertical Reynolds Stress (VRS) and the Vertical Buoyancy Flux (VBF) as:

$$HRS = -\overline{\mathbf{u}'v'} \cdot \partial_y \overline{\mathbf{u}} - \overline{\mathbf{u}'u'} \cdot \partial_x \overline{\mathbf{u}}; \tag{6}$$

$$VRS = -\overline{\mathbf{u}'w'} \cdot \partial_z \overline{\mathbf{u}}; \tag{7}$$

$$VBF = \overline{w'b'}; \tag{8}$$

where $\bullet'$ denote the deviation from the time average, $\overline{\bullet}$ the time average, $\mathbf{u}$ the horizontal velocity, $w$ the vertical velocity and $b = -g\frac{\rho}{\rho_0}$ the buoyancy.

Locally, the vertical buoyancy flux is maximum over the sloping topography (see Figure 5a) where the bathymetry is about 300 m (see Figure 5c). Indeed, it corresponds to the depth where the PV gradient sign changes occurs (see Figure 5b). The PV gradient sign changes originate from the sloping topography. In the upper part of the water column, the meridional gradient of PV is positive while it is negative in the lower part (see Figure 5b). This is a necessary condition for the baroclinic instability to occur. Most of the submesoscale eddies produced via baroclinic instabilities are observed below $300\,\mathrm{m}$ depth. Indeed, in the top 100 m, the stratification is stronger than below (see Figure 1c), so that it prevents the growth of baroclinic instabilities as discussed by Hetland (2017) and Wenegrat et al. (2018).

We also observe topographic Rossby waves propagating eastwards over the southern sloping topography and westwards over the northern one. In Figure 5d, we show the Hovmöller diagram of the potential vorticity anomaly at $500\,\mathrm{m}$ depth near the southern boundary of the domain. The potential vorticity anomaly is computed as:

$$Q' = Q - Q_{rest}, \tag{9}$$

with

$$Q_{rest} = f_0 \partial_z \overline{b}, \tag{10}$$

where $\overline{b}$ is the background vertical stratification and,

$$Q = (f_0 + \partial_x v - \partial_y u)\partial_z b - \partial_z v \partial_x b + \partial_z u \partial_y b \,, \tag{11}$$

the Ertel Potential Vorticity.

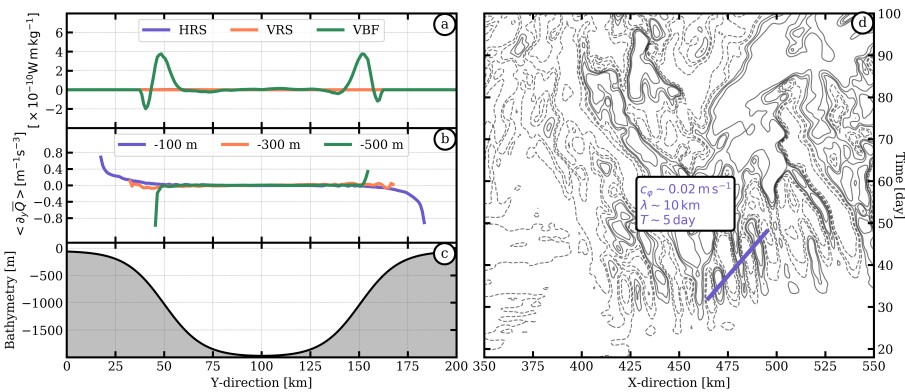

**Figure 5.** Diagnostics computed from NO-BBL. (a) Energy conversion terms: (blue) HRS, (orange) VRS and (green) VBF vertically integrated between 150 and 1000 m depth. (b) Timely and zonally averaged meridionnal PV gradient computed at (blue) 100, (orange) 300 and (green) 500 m depth. (c) Bathymetry. (d) Hovmoller diagram of potential vorticity anomaly at 500 m depth and $y \sim 40\,\mathrm{km}$

The solid blue line in Figure 5d indicates that submesoscale eddies propagate at about $0.02\,\mathrm{m\,s^{-1}}$. We compare this value with that obtained considering the propagation of topographic Rossby waves in a homogeneous fluid (the stratification is weak at $500\,\mathrm{m}$ depth). As defined in Cushman-Roisin and Beckers (2011), the dispersion relation of topographic Rossby waves in a homogeneous fluid leads to the following expression for the phase speed:

$$c_\varphi = \frac{\alpha_0 g}{f_0} \frac{1}{1 + \frac{gH}{f_0^2} k^2}. \tag{12}$$

With $\alpha_0 = 0.15$, $g = 9.81\,\mathrm{m\,s^{-2}}$, $f_0 = 6 \times 10^{-5}\,\mathrm{s^{-1}}$, $H = 1500\,\mathrm{m}$ and $\lambda = \frac{2\pi}{k} = 10\,\mathrm{km}$ (measured in Figure 5d), we obtain $c_\varphi \sim 0.02\,\mathrm{m\,s^{-1}}$ which is in agreement with that measured. Thus, submesoscale eddies are generated through baroclinic instabilities and propagate zonally under the influence of topographic Rossby waves. Then, submesoscale eddies are slowly advected off the coast by mesoscale eddies since the velocity field is very weak at $500\,\mathrm{m}$ depth compared with that at the

10 surface.

### 3.3.2 Frictional layer detachment

As mesoscale eddies drag over the sloping topography, they intensify the velocity shear within the bottom boundary layer leading to a topographic vorticity generating filament-like structures. This bottom boundary layer vorticity is directly linked with the use of a bottom drag and the bottom KPP and is possible only in BBL and BBL-CAPE. Subsequently, detached

15 vorticity filaments become unstable under the influence of horizontal shear. In Figure 6, it clearly appears that horizontal shear instabilities extract kinetic energy from the mean flow through the Horizontal Reynolds Stress ($HRS \gg VBF \gg VRS$). Shear instability is important near the topographic slope. Consequently, filaments roll up into small eddies. This also means that kinetic energy is transferred from larger to smaller scales. Since the bottom boundary layer is regularly destabilized, small eddies are produced by this frictional mechanism on both side of the basin.

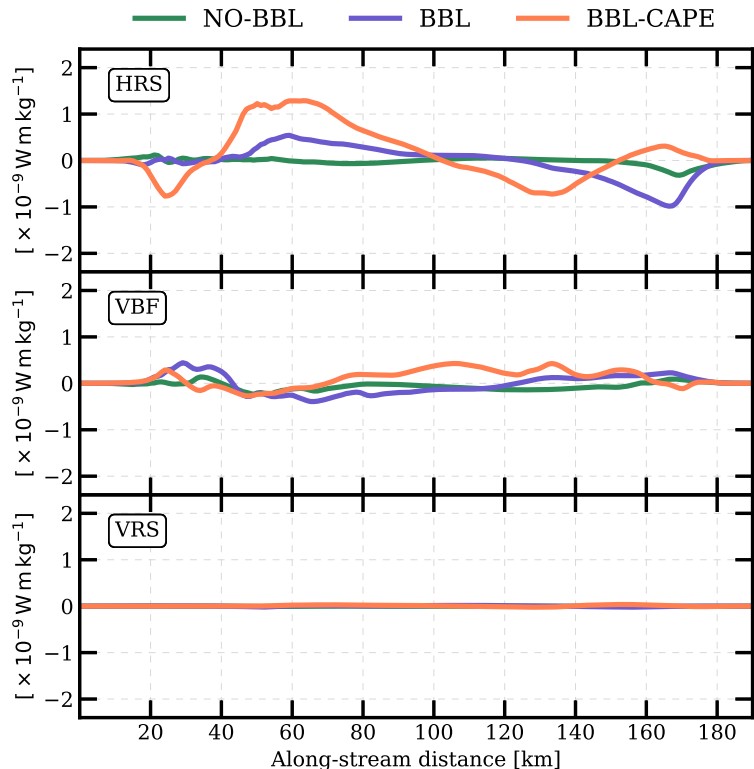

**Figure 6.** (Top) Horizontal Reynolds Stress, (Mid.) Vertical Reynolds Stress and (Bot.) Vertical Buoyancy Flux integrated between 100 and 300 m depth in the along-filament direction.

## 3.4 Structure and Lifecycle of the submesoscale eddies

In this section, we focus on the submesoscale eddies produced by frictional vorticity generation as the spatial scale and dynamical structure are in accordance with that found *via* the *in situ* measurement analysis.

Figure 7 shows the horizontal and vertical structures of such submesoscale anticyclone (a) and cyclone (b) found between 100 and 300 m depths in BBL experiment.

The submesoscale eddies are typically about 20 km wide and 150 m thick. The structure of the submesoscale eddies produced throughout the mesoscale eddy/topography interaction is consistent with the structure observed in the *in situ* measurements. The horizontal and vertical profiles of PV anomaly are close to Gaussian in both directions. The absence of sign reversal of the PV gradient is a clue to the stability of these small eddies (in the absence of external strain), hence their potentially long time life. Indeed, the background mesoscale shear is about $5 \times 10^{-6} \, \mathrm{s}^{-1}$. It corresponds to $\sim 10\% \times \zeta_{SCV}$ which is close to the quasi-geostrophic value necessary for vortices to elongate irreversibly.

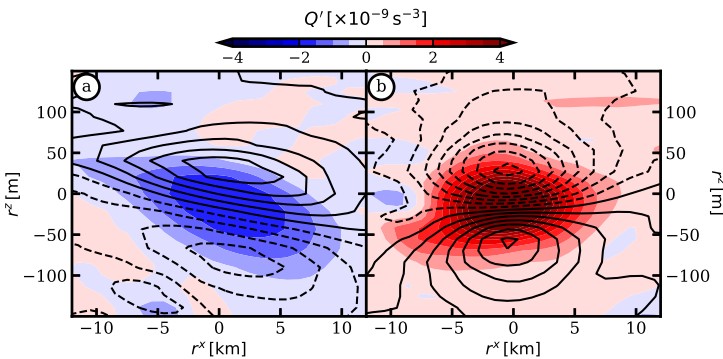

**Figure 7.** Vertical sections of potential vorticity anomaly for (a) an anticyclonic and (b) a cyclonic submesoscale eddies generated due to frictional effect (BBL experiment). Density anomalies associated with submesoscale eddies are shown in black contours.

The polarity of the submesoscale eddies is opposite to the polarity of the mesoscale eddy interacting with the sloping topography through the bottom boundary layer. In this paragraph, we describe the lifecycle of two submesoscale eddies: a cyclone and an anticyclone (see Figure 8 and 9). This lifecycle is typical of the submesoscale eddies observed in the model simulations, generated in the bottom boundary layer by interaction of the mesoscale eddies with the topography. Once formed, the subme-
soscale eddies rapidly merge with their neighbors (see Figure 9b, from day 17 to 49). The inverse energy cascade associated with the merging events tends to set larger scales than the bottom boundary layer scale energy injection scale. This process allows them to grow and provisionally, to withstand destruction induced by the mesoscale velocity shear. While they grow, they are also advected offshore *via* a dipolar coupling with their parent mesoscale eddy. Thus, they grow to a diameter of 15-20 km, comparable with that shown in Figure 2b. At this stage, their radius is still smaller than the deformation radius and $|\zeta^z/f| \geq 1$;
they are termed Submesoscale Coherent Vortices (SCV) (McWilliams, 1985).

In turn, as these vortices encounter the topographic slope, even smaller, opposite-signed vortices are produced (see Figure 8b at day 139 and Figure 9b at day 97); this leads to a slight dissipation of their energy. Then, the initial SCVs couple with the new opposite-signed SCVs, propagate again, and undergo straining. They are also sheared by the large vortices and thus
they decay in intensity (see for instance Figure 9b day 113). All these processes, generation of smaller vortices at the slope and vorticity erosion, contribute to feed the small scales in the enstrophy spectra.

The final evolution differs for the two SCVs. The cyclonic SCV merges with a mesoscale cyclone. The very details of this vortex interaction may not be fully captured by this hydrostatic model. A higher resolution, nonhydrostatic model would be
necessary to capture all the details of this small-scale vortex merger.[1] The anticyclonic SCV becomes less energetic after traveling over a long distance, and finally, it interacts again with the sloping topography. There, it loses most of its energy and

---

[1]This will be the subject of a forthcoming study.

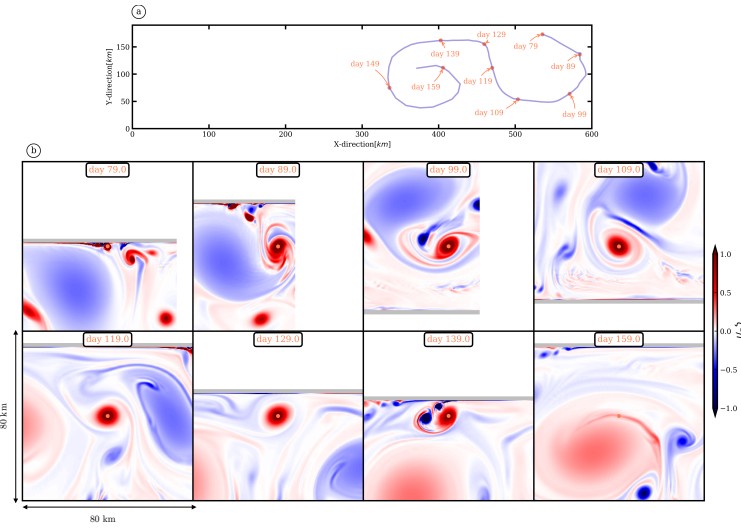

**Figure 8.** (a) Pathway of a submesoscale cyclone originates from the bottom boundary layer. (b) Major events from the birth till the death of the small vortex.

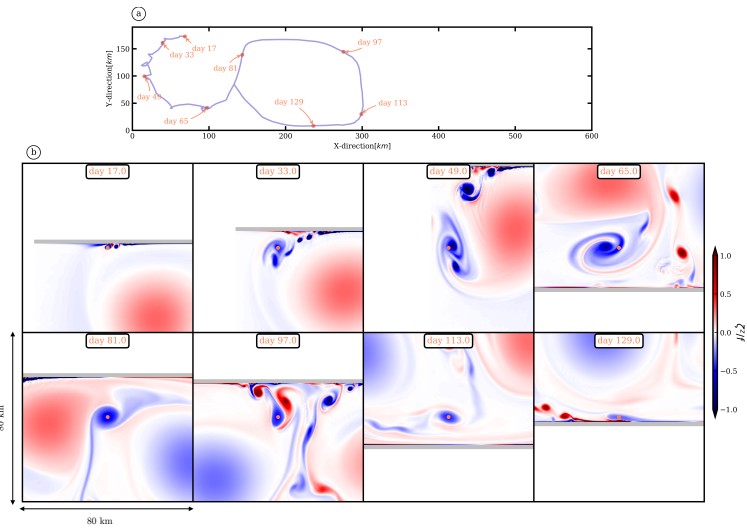

**Figure 9.** (a) Pathway of a submesoscale anticyclone originates from the bottom boundary layer. (b) Major events from the birth till the death of the small vortex.

disappears.

Depending on the number of merging events, their distance to the mesoscale eddies, and their interaction with the sloping topography, the SCVs lifetimes vary between 10 and 100 days and the distance that they cover is at most $600\,\mathrm{km}$. The global

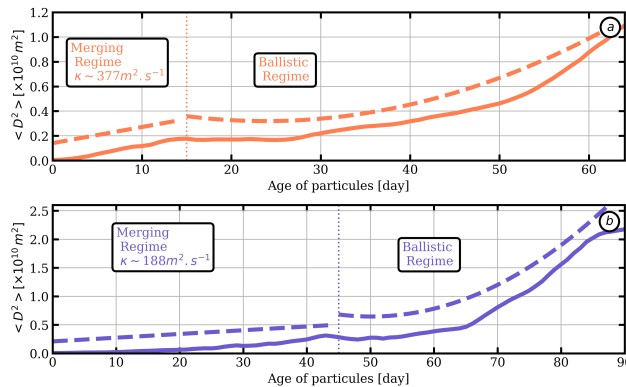

**Figure 10.** (a) and (b) time evolution of the mean square distance between particles and their center of mass regarding the submesoscale cyclone and anticyclone respectively (solid lines). Fits of merging and ballistic regimes are shown in dashed lines.

trajectory of these small eddies is essentially governed by the mesoscale flow. But their life cycle and the major events affecting them crucially depend on the population of submesoscale eddies and on the interaction with topography.

### 3.5 Spreading of outflow waters

We examine the capability of the submesoscale vortices to carry a tracer, such as the anomalously high salt content associated with the Persian Gulf Water and the Red Sea Water. To do so, we release about 10,000 Lagrangian particles into the two SCVs of Figures 8 and 9, at their birth (i.e. at day 79 for the cyclone and at day 17 for the anticyclone).

Figure 10 shows the mean squared distance between particles and their center of mass ($< D^2(t) >$) *vs* the age of particles. The results exhibit two regimes. The first regime is associated with a fairly linear trend of the root mean square distance with

time. For short times, a dispersion coefficient $\kappa$ can be found *via* :

$$< D^2(t) >= 4\kappa t, \tag{13}$$

as proposed by LaCasce and Bower (2000).

The diffusivities for the cyclonic and the anticyclonic SCVs are $377\,\mathrm{and}\,188\,\mathrm{m}^2\mathrm{s}^{-1}$ respectively. Our estimate corresponds to the dispersion associated with the SCV merging events. During this period, the SCVs lose only a few particles compared to

the second regime. This second regime exhibits an enhanced particle loss. This stage corresponds to the dipolar advection of submesoscale eddies coupled with their mesoscale neighbor. This regime is characterized as ballistic with $< D^2(t) >\propto t^2$.

Through time, we calculate the number of particles located within 8 km of each submesoscale eddy. In Figure 11a, we show the trajectories of submesoscale eddies in solid lines and the positions of selected particles with dots. The time evolution of

the Probability Density Function (PDF) of the normalized relative vorticity of the selected particles is shown in Figure 11b,c.

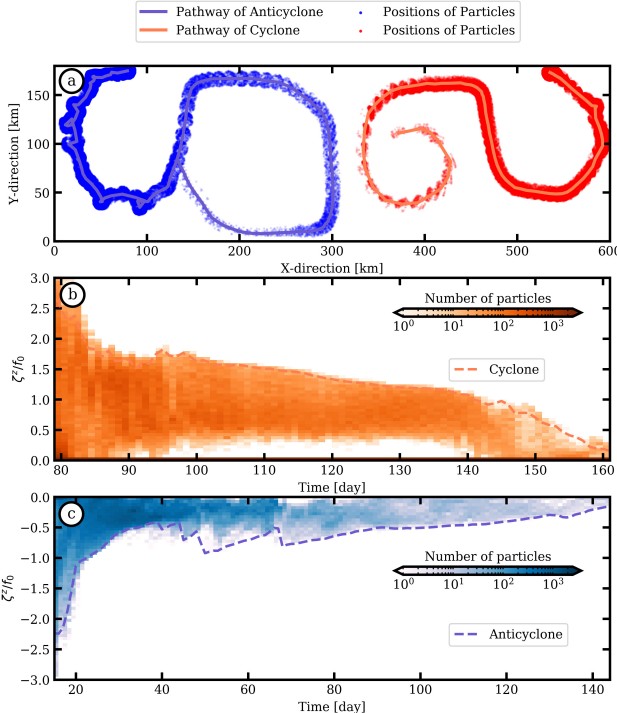

**Figure 11.** (a) Trajectories of the small vortices considered in Figure 8 (orange line) and 9 (blue line) (blue) and positions of particles located at a distance smaller than 8 km from the smaller vortex. (b) and (c) Time evolution of the histogram of vertical relative vorticity normalized by the planetary rotation of particles (colors) and the normalized vertical relative vorticity associated with the small vortices (dashed lines).

The normalized relative vorticity associated with the submesoscale eddies dramatically decreases during the first days. This corresponds to the merging events after which the submesoscale eddy is larger in size but is less intense (note that only potential vorticity is nearly conserved, in the limit of small viscosity). This relative vorticity decrease occurs via filamentation and spatial re-organization during the merging process; this re-organization is associated with relative vorticity to vortex stretching conversion, with an increase in potential energy (Ciani et al., 2016). The number of particles trapped in each eddy varies during the merging events. As the submesoscale eddies rotate around their parent mesoscale eddy, they are sheared and they lose particles at their periphery.

Finally, the impact of the submesoscale eddy production due to bottom frictional effect on the diffusion of particles is highlighted by the comparing three experiments (i.e.: NO-BBL, BBL and BBL-CAPE). Particles are initially seeded at the southwest edge of the channel between 100 and 300 meter depths. Figure 12 shows the positions of particles with time. In NO-BBL experiment, without bottom boundary layer vorticity generation (Figure 12, 1$^{\text{st}}$ column), particles are advected mostly by mesoscale eddies. Particles remain mostly in the left hand side of the domain. In BBL and BBL-CAPE experiments

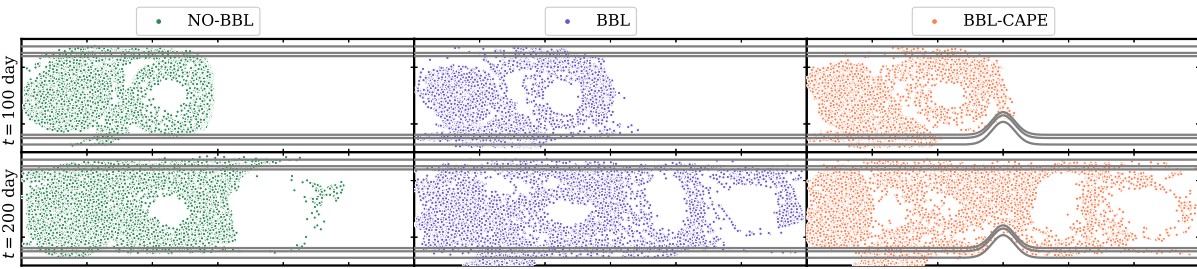

**Figure 12.** Positions of particles for the three experiments at days 100 and 200.

(Figure 12, 2$^{\text{nd}}$ and 3$^{\text{rd}}$ columns respectively), particles are advected by filaments over the sloping topography, subsequently by submesoscale eddies, and by mesoscale eddy as well. The dispersion of particles is then more efficient in BBL and BBL-CAPE experiments than in NO-BBL experiment.

Following LaCasce (2008), we can estimate an equivalent diffusivity coefficient through time as:

$$\kappa = \frac{1}{2} \frac{\mathrm{D}}{\mathrm{D}t} < D^2 > . \tag{14}$$

In Figure 13, we show the time evolution of the relative dispersion $< D^2 >$ where $D$ is the distance between pairs of particles and $< \bullet >$ is the ensemble average over all the pairs of particles. During the first 25 days of integration, the relative dispersion of particles is similar in the three experiments. Later, the relative dispersion increases more strongly in BBL and BBL-CAPE experiments. Particles trapped within submesoscale eddies can reach the cyclones and anticyclones on the right side of the domain more easily and travel longer distances. The cape does not play a significant role in the particle dispersion due to the initial configuration of the mesoscale eddies. The estimations of the dispersion coefficient yield $\sim 1000\,\mathrm{m^2 s^{-1}}$ in both experiments while in NO-BBL experiment it is about $700\,\mathrm{m^2 s^{-1}}$.

In Figure 14a, we compare the number of particles transferred from the left to the right hand side of the channel. In BBL and BBL-CAPE experiments, the amount of particles lying in the right hand side of the channel is $\sim 15\%$ of the total number of particles. This amount of particles decreases to $\sim 5\%$ in NO-BBL experiment. In BBL experiment, the amount of particles oscillates through time as a result of the recirculation of particles due to the mesoscale cyclone lying in $x \in [300, 400]\,\mathrm{km}$. These oscillations do not appear in BBL-CAPE experiment which means that the cape does not allow particles to recirculate.

In Figure 14b,c, we show the PDF of the relative vorticity normalized by the planetary rotation associated with particles at day 200 for particles lying in the left and right side of the channel respectively. In the three experiments PDFs are Gaussian shaped and have zero bias, meaning that the influence of cyclonic and anticyclonic motions is similar. However, on the left hand side of the channel, particles can be trapped in the turbulent bottom boundary layer and remain in it for long time. This can be seen in Figure 12 in BBL and BBL-CAPE experiments at day 200. There, the bottom boundary layer vorticity is positive and large (i.e.: $|\zeta^z / f_0| > 1$.). That is why, PDF associated with BBL and BBL-CAPE experiments show that particles can have very large value of relative vorticity. On the other hand, in BBL and BBL-CAPE experiments, regarding the particles lying in the right hand side of the channel, PDFs flatten and large relative vorticity values are reached (i.e.: $|\zeta^z / f_0| > 0.5$),

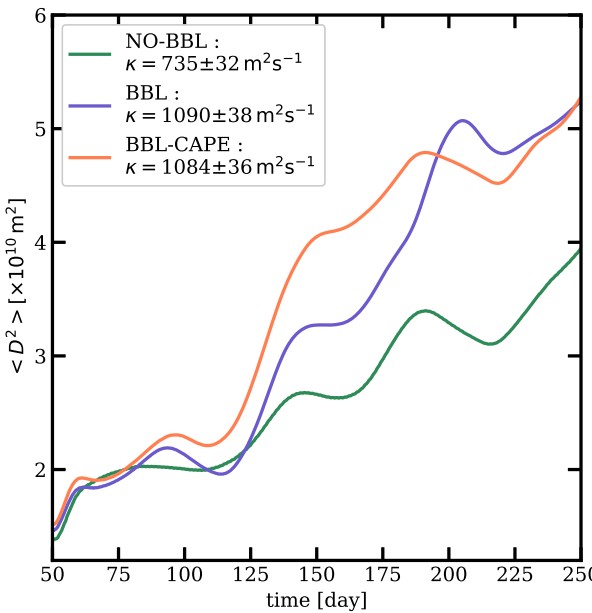

**Figure 13.** (a) Time evolution of the dispersion $<D^2>$ computed for the three experiments: (green) NO-BBL, (blue) BBL and (orange) BBL-CAPE.

then the variances increase as a signature of the particle trapping in submesoscale eddies. This result highlights the impact of submesoscale eddies on the pathways and spreading of particles. Without subsurface submesoscale eddy generation due to bottom frictional effect, the fate of particles is mostly driven by the mesoscale eddy field. In this case, particles are trapped in mesoscale eddies or recirculate at the periphery of mesoscale eddies. However, with the subsurface submesoscale eddy generation by bottom frictional effect, particles can be trapped and advected by submesoscale eddies. Then, they can travel over large distance and long time, see also Vic et al. (2018) on the role of SCVs in the spreading of tracers.

## 4    Conclusions

*In situ* observations have shown filaments and small eddies of Persian Gulf Water, formed from its outflow along the continental slope of the Gulf of Oman. At this time, a row of alternate signed mesoscale eddies was present in the gulf. We used a primitive-equation numerical model to represent the ambient mesoscale circulation in an idealized rectangular basin. Specifically, the circulation is dominated by the interaction of a row of surface intensified mesoscale eddies with topographic slopes. We showed that the friction of these vortices on the slope (in a slanted bottom boundary layer) is the primary mechanism for the generation of opposite-signed relative vorticity on the slope. The vorticity filaments thus created then undergo shear instability and form small eddies. These small eddies merge and grow in size. They also pair (cyclones with anticyclones) and propagate away from their region of formation. At the surface, these small eddies have a signature and can thus be detected via

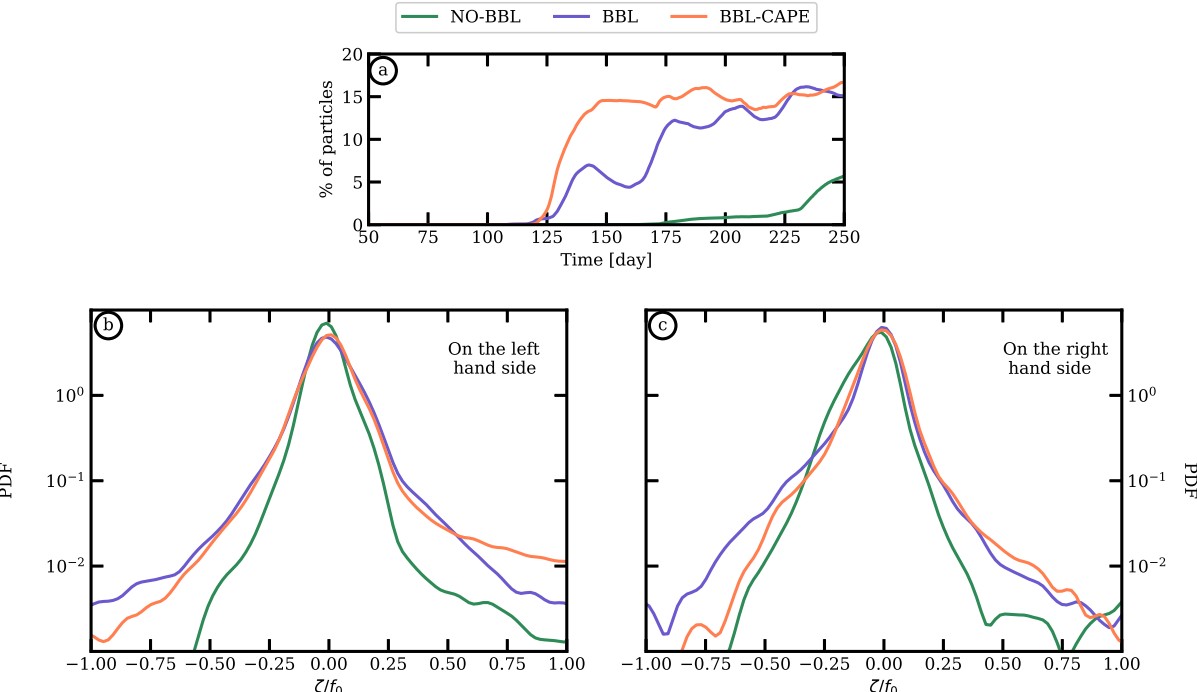

**Figure 14.** (a) Time evolution of the percentage of particles lying in $x \in [400; 600]$ km the three experiments: (green) NO-BBL, (blue) BBL and (orange) BBL-CAPE. (b,c) Probability Density Function of the relative vorticity normalized by the planetary rotation associated with particles on the left ($x < 300$ km) and right ($x > 300$ km) hand-side of the domain regarding (green) NO-BBL, (blue) BBL and (orange) BBL-CAPE.

satellite altimetry. This is due to their strong intensity and to the shallow depth at which they lie. Also, below $300$ m depth, the stratication is weak and baroclinic instability can develops and creates turbulence. The submesoscale eddies propagate zonally under the influence of topographic Rossby waves.

5  Newly formed submesoscale eddies are advected along the rim of ambient mesoscale eddies, and eventually interact with the seafloor topography, in turn generating smaller-scale eddies. This is manifested by a shallower slope of the enstrophy spectrum (more energetic small scales of vorticity). Finally, these small eddies can be destroyed by the shear of the mesoscale eddies (or be incorporated into them) or crash onto the topographic slope and be eroded there. Their lifetime is usually shorter than $3$ months and the length of their pathway is at most $600$ km .

We have also shown that these small eddies play a non negligible role in the transport and dispersion of tracers (which can be salinity in the Gulf of Oman) by comparing two simulations with and without the bottom boundary layer vorticity generation. Due to submesoscale eddies generated by frictional effect, particles can travel over large distance more rapidly. Finally, the

presence of a cape, such as the Cape of Ra's al Hamra in the Gulf of Oman, does not impact significantly the diffusion of a passive tracer such as PGW.

This study raises other worthwhile questions. In particular, the initial configuration was designed to be representative of the sloping topography as well as the mesoscale eddies found in the Gulf of Oman. In the Gulf of Aden, the slope of the topography is different and, mesoscale eddies can reach $1000\,\text{m}$ depth. Both aspects should not have an impact on the mechanism generating submesoscale eddies but rather on the location of their generation thus on the dispersion of the Red Sea outflow Water.

Also, we will use a non hydrostatic model to study the interaction between two SCVs or between one SCV and one mesoscale eddy in greater detail. The larger upwelling of nutrients in such a study will be important for onset of algae blooms. Our simulations were done without any surface forcing and generated fewer filaments and eddies. In the real ocean, filaments and eddies arise at a range of scales due to turbulent interactions and from instability of upwelling fronts. The interaction between surface and subsurface SCVs in the vertical is another aspect that calls for further investigations.

*Acknowledgements.* We acknowledge support from ANR ASTRID Maturation project DYNED ATLAS and from UBO. We thank CNRS and RFBR for support under PRC project 1069 (in French classification) and 16-55-150001 (in Russian classification). Mikhail Sokolovskiy was supported also by the Ministry of Education and Science of the Russian Federation (Project No.14.W.03.31.0006)

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
