# Peer review of "The life cycle of submesoscale eddies generated by topographic interactions"

_Ocean Science, 2019_

## Referee Comment (RC1) · Anonymous Referee #1 · 21 May 2019

In this manuscript the authors investigate the role of submesoscale eddies, generated along sloping topography, in an idealized version of the Arabian Sea. A reentrant channel with sloping sidewalls is initialized with an array of mesoscale eddies, and the evolution of submesoscale eddies in the interior are examined. A portion of the manuscript is motivated around the role that submesoscale eddies may play in the dispersal of salty water from the Gulf of Oman, accomplished using offline particle tracking. Overall this is a well written and interesting manuscript, although there are several points listed below that I would ask the authors to consider.

[Figure]

**1 Major Comments**

1. Section 3.3.1 needs some clarifying and expansion. First, as I understand the section and figure 5, you are simply arguing that the submesoscale eddies at 500 m depth are generated via baroclinic instability. This seems plausible, but should be put in the context of recent work on the topic (eg. Hetland 2017, and Wenegrat et. al 2018). Likewise, it was not clear to me whether the focus throughout on the mechanism being a topographic Rossby wave was meant to distinguish this in some way from the basic baroclinic instability mechanism over a slope (in which case this needs clarification), or whether it was just a particular way of introducing why baroclinic instability can happen over a slope (which I would argue is unnecessarily complicated and could just be replaced throughout by 'baroclinic instability').
It would also be good to dig a bit deeper in this section into related questions such as:

   - Why is this mechanism not generating as active an eddy field at shallower depths in EXP1? A possible explanation might be the dependence of the instability on the Slope Burger number, such that the stronger stratification at shallower depths suppresses growth (Wenegrat et al. 2018).
   - Is the instability trapped between the bottom and the pycnocline? Ie. what sets the vertical scale?
   - What determines the separation of the eddies off the topography into coherent vorticies?

2. The motivation of the study mentions both the Persian Gulf and Red Sea outflows, however the study is really only focused on the 200 m depth range (ie. the Persian Gulf water). For instance, both the detailed case studies and the particle tracking are focused only on the 100-300 m depth range. This choice may reflect the fact

that it is only in this depth range where there are substantial differences between the experiments. However, the most active submesoscale eddy field is at 500 m depth (eg. figure 3).

As such, I would suggest that the particle tracking analysis should be repeated for 500 m depth. While there are likely not significant differences between experiments at this depth, the findings would have implications for the accuracy of lower-resolution models in capturing the spread of Red Sea water.

3. The differences in particle dispersion between the 3 experiments are being attributed to the submesoscale eddies in the interior. However, an alternate hypothesis would be that the differences arise due to boundary layer dynamics (absent in EXP1). Histograms are shown for vorticity field sampled by the particles (eg. figure 13 b, c, d), showing heavier tails in EXP2 and EXP3, which is interpreted as evidence of the role of submesoscale eddies. However, this same sort of pattern could also occur if the particles were randomly sampling the underlying flow field (which would have a heavier tailed vorticity distribution in EXP2 and EXP3). A bit more analysis of this section would make the argument for the role of submesoscale eddies more convincing. For example, one could look at the changes in the particle sampled vorticity distribution relative to the changes in the underlying distribution across the whole domain. You could also try comparing distributions between particles which make it to the right-hand side of the domain to those that don't.

**2 Minor Comments**

1. You have high resolution in the vertical (100 $\sigma$ levels), and moderately high-resolution in the horizontal, with moderately steep topographic slopes. Are hydrostatic pressure gradient errors a concern for this setup? It would be good to

comment on this (eg. grid stiffness etc) in section 2.2.

2. You should comment a bit more on the choice to model the dispersion of dense water as a passive tracer. As I understand the setup, really what you are intending to say is that the passive particles are meant to act as a proxy for high-salinity water. I assume that this choice was made because introducing a salinity gradient in the initial condition would be problematic with the re-entrant domain.

3. Given that you only have 3 runs, I would suggest renaming them with more informative names, which is very helpful to the reader. For example, you could choose to name them NO-BBL, BBL, and BBL-CAPE, or any other variant that immediately conveys the setup.

4. In section 3.5 you introduce two different definitions of the diffusivity (equations 13 and 14). Please clarify why these two definitions are given, why they don't agree, and if possible clean this section up a bit by using only one.

5. The final paragraph of the manuscript feels out of place, and not well supported. For example 'the vertical motions then are of importance to the uplift of nutrients in the ocean and then onset of algae blooms' is extremely speculative when considering an instability at 200 m depth. As this paragraph really is just laying out a variety of future work the authors plan to carry out, it is not entirely relevant to the bulk of the manuscript, and I'd suggest removing it.

6. In some of the figures the subplots lack labels/scales on the axes. For instance, figure 8 shows an eddy in plan view, without axes labels. The moving focus region between subplots would make it hard to label with absolute position, however you could at least add some scale to the x-y axes (ie are the subplots showing a 10km x 10 km region? 100 km x 100 km?).

7. Figure 10: Describe the meaning of the dashed lines in the caption.
8. The wording in the abstract connecting the findings here to the Persian Gulf Water and Red Sea water is a bit too strong. I would suggest rewording to: *...and their potential impact on the spread of Persian Gulf Water..'* and *'This shows the potentially important role of submesoscale eddies...'.*

9. Spelling: 'wit*h*out' near line 15 in the abstract.

10. I assume that the black contours in Figure 2 (b) and (c) are density, however this is not indicated in the caption.

**References**

Hetland, R.D., 2017: Suppression of baroclinic instabilities in buoyancy-driven flow over sloping bathymetry. *J. Phys. Oceanogr.*, **47**, 49-68.

Wenegrat, J.O., J. Callies, and L.N. Thomas, 2018: Submesoscale baroclinc instability in the bottom boundary layer. *J. Phys. Oceanogr.*, **48**, 2571-2592.

---

## Short Comment (SC1) · 29 May 2019

In the present manuscript, authors study possible mechanisms of the submesoscale generation in the Gulf of Oman and the Gulf of Aden. They indicate on two mechanisms: the arrested Rossby waves and the frictional generation vorticity in the bottom layer and horizontal shear instability. Authors supposed that the submesoscale dynamics can be responsible for the spreading of the Persian Gulf Water and Red Sea Water. Based on the idealized numerical simulations, authors study comprehensively both mechanisms and indicate that the frictional mechanism is more effective than the arrested Rossby waves to the generation of submesoscale variability. The presented results no doubt are useful and give rise new understanding to the spreading the Persian Gulf Water and Red Sea Water. However, there are some shortcomings that need

to correct.

The major shortcoming is a poor comparing the simulation results with natural measurements. How to extent the considered mechanisms can characterize the spreading the Persian Gulf Water and Red Sea Water? As well as, authors do not compare their results with that from the other numerical simulation studies. The model configuration description is very brief. Please specify the used parameterizations of subgrid-scale, open boundary conditions, momentum fluxes, heat and salt fluxes on the upper boundary of the channel. Please compare the observed temperature and salinity profiles with these from the numerical simulations. What differences between them exist? The generation period of intensive submesoscale variability is about 200 days. During this period the background conditions: the spatial structure of mesoscale eddies and temperature and salinity can change. Please give proof the stability of the mesoscale eddy row during 200 days.

Short comments 1. Line 15. Please change, Figure 5a on Figure 5d. 2. Please give the definition of the SCV.

---

## Referee Comment (RC2) · Anonymous Referee #2 · 15 Jun 2019

In this idealized numerical study inspired by inset observations of the mesoscale eddies in the Gulf of Oman, the authors use a submesoscale resolving model to investigate the formation and life cycle of submesoscale vortices and their impact on the Persian Gulf water and the Red Sea water.

The geometry is idealized to parallel north and south walls with topographic slopes, with a row of mesoscale vortices in between. The initial condition is inspired by a Physindien11 observational campaign.

Two mechanisms are explored for the generation of submesoscale vortices, both related to the flow interacting with the topographic slope. The first mechanism is frictional vorticity generation in BBL, and second due to topographic Rossby waves breaking when a mesoscale anticyclone interacts with the topographic slope. Submesoscale

eddy lifecycles are discussed with several numerical examples, and their impact on particle dispersion within the Gulf is also discussed. I enjoyed reading sections 3.3, 3.4 and 3.5 and understanding the implications of submesoscale eddies to the dispersal of PGW.

There is some evidence shown into sections (2.1 and 3.1, and Figure 2) to motivate the numerics, but in the rest of the paper, there is no inter-comparison with the observational data.

The presentation of results in the paper is convincing though I would like the authors to address the following questions before I can recommend publication:

1. Section 2.1 and 3.1: Did the Physindien 2011 campaign show evidence of submesoscale flows? While there is some evidence in the GO13 (figure 2c) section to support the filaments of Persian Gulf Water flow in T and S, but there is no velocity information presented from the corresponding ADCP sections. I would like the authors to present some velocity information from the observational campaign in section 2. What scales do you see (after suitable averaging in the ADCP section? The accuracy of horizontal velocity components is mentioned, but no data is shown from the ADC section so this is superfluous information. 3. Do you see any evidence of submesoscale vortices in the velocity structure from the observations? At what depths? How high were the 2-d vorticity that you could observe in the ADCP sections? At what depth?

4. What do the Observed KE spectra show at various depths for the sections shown in Figure 2. In the ranges that overlap — how do the spectra from the model and the observations compare (in terms of slopes, etc.?)?

5. Page2, Lines 27-28: The simulations you are doing are very idealized and you are using the observations as an inspiration, so saying that the simulations "specifically designed to resemble the local geography of the Gulf of Aden and the Gulf of Oman" is incorrect. The geometry is very idealized, and the Gulf is variable in width unlike your simulations and has many other geographical details. You need to modify this

sentence to reflect this.

6. Page 3, Line 17: The accuracy of ADCP u,v is mentioned to be 0.5cm/s. With what averaging? No ADCP section is shown, and no averaging information is mentioned, so this information is not useful to your reader.

7. Figure 2a). Change the color for the GO13 section, it is very hard to see the line corresponding to GO13 in this panel.

8. Figure 3. We can see intense vorticity variations in this figure at sub-mesoscales. Does the density stratification vary on these scales? You should present either density or $N^2$ from EXP1,2 and 3 at these depths as well — are these vortices seen in the density structure as well?

9. Page 8Line 15, and Page 9 Line 1-5: We know from your initial conditions and from the observations that the flow is stratified. Why does the "homogeneous" fluid TRW match the observed phase speeds? What is the phase speed if stratification was taken into account?

10. Section 3.5 and 4: Particularly for the last part of section 4, the presentation of the text needs to be improved, and I have offered several suggestions in the attached annotated version of the ms.

Please also note the supplement to this comment:
https://www.ocean-sci-discuss.net/os-2019-3/os-2019-3-RC2-supplement.pdf

**Supplement:**

[Figure]

[Figure]

**The life cycle of submesoscale eddies generated by topographic interactions**

Mathieu Morvan[1], Pierre L'Hégaret[1], Xavier Carton[1], Jonathan Gula[1], Clément Vic[2],
Mikhail Sokolovskiy[3], and Konstantin Koshel[4]

[1]LOPS, Univ. Brest-CNRS-IFREMER-IRD, IUEM
[2]National Oceanographic Center, University of Southampton, European Way, Southampton, SO14 3ZH, UK
[3]Institute of Water Problems of the RAS, Ul Gubkina 3, Moscow, 199333, Russia, Shirshov Institute of Oceanology of RAS,
36 Nahimovskiy pr., Moscow, 117997, Russia
[4]V.I.Il'ichev Pacific Oceanological Institute, 43, Baltiyskaya Street, Vladivostok, 690041, Russia

**Correspondence:** Mathieu Morvan (mmorvan3@univ-brest.fr)

**Abstract.**

The Persian Gulf Water and Red Sea Water are salty and dense waters recirculating at subsurface in the Gulf of Oman and the Gulf of Aden respectively, under the influence of mesoscale eddies which dominate the surface flow in both semi-enclosed basins. In situ measurements combined with altimetry indicate that the Persian Gulf Water is driven by mesoscale eddies in the

5 form of filaments and submesoscale structures. In this paper, we study the formation and the life cycle of intense submesoscale vortices and their impact on the spread of Persian Gulf Water and Red Sea Water. We use a three-dimensional hydrostatic model with submesoscale-resolving resolution to study the evolution of submesoscale vortices. Our configuration is an idealized version of the Gulf of Oman and Aden: a zonal row of mesoscale vortices interacting with north and south topographic slopes. Intense submesoscale vortices are generated in the simulations along the continental slopes due to two different mech-

10 anisms. The first mechanism is due to frictional generation of vorticity in the bottom boundary layer, which detaches from the topography, forms an unstable vorticity filament, and undergoes horizontal shear instability that leads to the formation of submesoscale coherent vortices. The second mechanism is inviscid and implies arrested topographic Rossby waves breaking and forming submesoscale coherent vortices where a mesoscale anticyclone interacts with the topographic slope. Submesoscale vortices subsequently drift away, merge and form larger vortices. They can also pair with opposite signed vortices and travel

15 across the domain. They can weaken or disappear via several mechanisms, in particular fusion into the larger eddies or erosion on the topography. Particle patches are advected and sheared by vortices and are entrained into filaments. Their size first grows as the square root of time, a signature of the merging processes, then it increases linearly with time, corresponding to their ballistic advection by submesoscale eddies. On the contrary, witout intense submesoscale eddies, particles are mainly advected by mesoscale eddies; this implies a weaker dispersion of particles than in the previous case. This shows the important role of

20 submesoscale eddies in spreading Persian Gulf Water and Red Sea Water.

[Figure]

**1 Introduction**

Mesoscale eddies [$\mathcal{O}$(10-100)km] are ubiquitous in the world's oceans. The mesoscale eddy field dominates the energy content of oceanic currents at sub-inertial frequencies and is crucial to large-scale budgets of heat and geochemical tracers. Although their generation mechanisms are clearly identified and quantified, largely stemming from large-scale current instabilities, how

5   they dissipate their energy remains unclear (Ferrari and Wunsch, 2009). The interaction of these mesoscale vortices can lead to the formation of smaller scale features (Carton, 2001). Submesoscale eddies [$\mathcal{O}$(1-10)km] can be generated as the result of frontal instabilities (Capet et al., 2008a, b, c) feeding off the energy of the mesoscale currents. Recently, numerical models highlighted one possible mechanism for mesoscale energy dissipation, which is the interaction of mesoscale eddies with the underlying seafloor topography (Molemaker et al., 2015; Gula et al., 2015, 2016; Vic et al., 2015). This specific interaction

10  also leads to the formation of submesoscale eddies. In this regime, the influence of stratification is still important, but the Coriolis force is less prevalent in the horizontal momentum budgets (viz. the Rossby number of the flow is not always small; McWilliams (2016)). Submesoscale eddies have been sampled for decades, yet their lifecycle (from generation to dissipation) is poorly understood (McWilliams, 1985; D'Asaro, 1988; Bosse et al., 2015).

15  The Arabian Sea is home to an energetic mesoscale eddy field (Carton et al., 2012; Vic et al., 2014; L'Hégaret et al., 2015). In particular, they can interact with the salty and dense waters of the outflows from the Persian Gulf and from the Red Sea (Bower et al., 2002; Al Saafani et al., 2007). These salty and dense waters flow out of these seas into the Gulf of Oman and into the Gulf of Aden and settle at 250-300 m and 600-1000 m depths, respectively (Pous et al., 2004a; Bower and Furey, 2012). These two gulfs also receive Rossby and Kelvin waves, and vortices, propagating westwards from the Arabian Sea (L'Hégaret

20  et al., 2013). Mesoscale vortices divert the outflow paths away from the coast, advect them along curved trajectories around the eddy rims, elongate these outflows as salty filaments and finally, can break these filaments into small eddies (Pous et al., 2004b; Carton et al., 2012). The chaotic dispersion of a tracer by a mesoscale vortex row in a realistic numerical simulation of the Japan Sea has previously been studied by Prants et al. (2011). However, the details of the spreading mechanism, involving submesoscale eddies, has not been explored.

25

In this paper, we focus on the life cycle of the submesoscale eddies generated by mesoscale eddy-topography interactions, combining observational data from dedicated campaigns and numerical simulations specifically designed to resemble the local geography of the Gulf of Aden and the Gulf of Oman. 
[revised manuscript text omitted]

15  As mentioned in the previous section, two mechanisms implying eddy/topography interaction, leading to the formation of small vortices are at play.

[Figure]

[Figure]

**Figure 4.** Kinetic Energy (KE) spectra comparison of the three experiments: EXP1 (green), EXP2 (blue), EXP3 (orange). Spectra are computed at (a) 10, (b) 200 and (c) 500 meter depth at day 200. Black lines stand for benchmarks of power laws.

**3.3.1 Unstable Topographic Rossby Waves**

The first mechanism is an inviscid mechanism visible below $300\,\mathrm{m}$ in all experiments. This mechanism is related to the breaking of Topographic Rossby waves (TRW hereafter). TRW are triggered by the eddy velocity fields acting as the perturbation along the sloping bathymetry. This can be seen in EXP1, in the absence of frictional effects. In Figure 5d, we show the Hovmöller diagram of the potential vorticity anomaly at $500\,\mathrm{m}$ depth near the southern boundary of the domain. The potential vorticity anomaly is diagnosed and computed as:

$$Q' = Q - Q_{rest},\tag{6}$$

with

$$Q_{rest} = f_0 \partial_z \overline{b},\tag{7}$$

where $\overline{b}$ is the background vertical stratification and,

$$Q = (f_0 + \partial_x v - \partial_y u)\partial_z b - \partial_z v \partial_x b + \partial_z u \partial_y b \,,\tag{8}$$

the Ertel Potential Vorticity.

The solid blue line in Figure 5a indicates that the phase speed is about $0.02\,\mathrm{m\,s^{-1}}$. We compare this value with that obtained considering the propagation of TRW in a homogeneous fluid. As defined in Cushman-Roisin and Beckers (2011), the dispersion

[Figure]

[Figure]

[Figure]

**Figure 5.** Diagnostics computed from EXP1. (a) Energy conversion terms: (blue) HRS, (orange) VRS and (green) VBF vertically integrated between 150 and 1000 m depth. (b) Timely and zonally averaged meridionnal PV gradient computed at (blue) 100, (orange) 300 and (green) 500 m depth. (c) Bathymetry. (d) Hovmoller diagram of potential vorticity anomaly at 500 m depth.

*! Why does this work as well as it does?*

relation of TRW in a homogeneous fluid leads to the following expression for the phase speed:

$$c_\varphi = \frac{\alpha_0 g}{f_0} \frac{1}{1 + \frac{gH}{f_0^2} k^2}. \tag{9}$$

With $\alpha_0 = 0.15$, $g = 9.81\,\mathrm{m\,s^{-2}}$, $f_0 = 6 \times 10^{-5}\,\mathrm{s^{-1}}$, $H = 1500\,\mathrm{m}$ and $\lambda = \frac{2\pi}{k} = 10\,\mathrm{km}$ (measured in Figure 5d), we obtain $c_\varphi \sim 0.02\,\mathrm{m\,s^{-1}}$.

5    The TRW destabilizes below $300\,\mathrm{m}$ leading to the formation of submesoscale eddies. In Figure 5a, we show that TRW are baroclinically unstable over the sloping topography.

We characterize the instabilities by computing the energy transfer terms (Gula et al., 2016), i.e.: the Horizontal Reynolds Stress (HRS), the Vertical Reynolds Stress (VRS) and the Vertical Buoyancy Flux (VBF) as:

$$HRS = -\overline{\mathbf{u}'v'} \cdot \partial_t \overline{\mathbf{u}} - \overline{\mathbf{u}'u'} \cdot \partial_x \overline{\mathbf{u}}; \tag{10}$$

$$VRS = -\overline{\mathbf{u}'w'} \cdot \partial_z \overline{\mathbf{u}}; \tag{11}$$

$$VBF = \overline{w'b'}; \tag{12}$$

where $\bullet'$ denote the deviation from the time average, $\overline{\bullet}$ the time average, $\mathbf{u}$ the horizontal velocity, $w$ the vertical velocity and

15   $b = -g\frac{\rho}{\rho_0}$ the buoyancy.

[Figure]

[Figure]

[Figure]

**Figure 6.** (Top) Horizontal Reynolds Stress, (Mid.) Vertical Reynolds Stress and (Bot.) Vertical Buoyancy Flux integrated between 100 and 300 m depth in the along-filament direction.

Locally, the vertical buoyancy flux is maximum over the sloping topography. There, the bathymetry is about 300 m depth. Indeed, it corresponds to the depth where the PV gradient sign change occurs. The PV gradient sign change originate from the sloping topography. In the upper part of the water column, the meridional gradient of PV is positive while it is negative in the lower part (see Figure 5b). This is a necessary condition for the baroclinic instability. The TRW propagate eastwards

5   over the southern sloping topography and westwards over the northern one. So, over the sloping topography, TRW propagate in the same direction as the velocity field associated with the mesoscale cyclones. On the contrary, TRW propagate in the opposite direction of the current associated with mesoscale anticyclones. The TRW can be arrested and destabilizes when the velocity of the flow is opposite to its phase speed. This can happen only when the TRW encounters a mesoscale anticyclone.

10   **3.3.2   Frictional layer detachment**

As mesoscale eddies drag over the sloping topography, they intensify the velocity shear within the bottom boundary layer (BBL hereafter) leading to a topographic vorticity generation forming filament-like structures. This BBL vorticity is directly linked with the use of a bottom drag and the bottom KPP and is possible only in EXP2 and EXP3. Subsequently, detached vorticity filaments become unstable under the influence of horizontal shear. In Figure 6, it clearly appears that horizontal shear instabili-

15   ties extract kinetic energy from the mean flow through the Horizontal Reynolds Stress ($HRS \gg VBF \gg VRS$). Barotropic instability is important near the topographic slope. Consequently, filaments roll-up into small eddies. This also means that

[Figure]

[Figure]

**Figure 7.** (a) Horizontal and (b) vertical profiles of potential vorticity anomaly associated with two SCVs.

kinetic energy is transferred from larger to smaller scales. Since the bottom boundary layer is regularly destabilized, small eddies are produced regularly by this frictional mechanism at each coast. When these eddies form and detach from the coast, they adjust cyclogeostrophically.

**3.4 Structure and Lifecycle of the submesoscale eddies**

In this section, we focus on the submesoscale eddies produced by BBL vorticity generation as the spatial scale and dynamical structure are in accordance with that found *via* the *in situ* measurement analysis.

Figure 7 shows the horizontal (a) and vertical (b) structures of such submesoscale eddies found between $100$ and $300\,\mathrm{m}$ depths in EXP2.

The submesoscale eddies are typically about $20\,\mathrm{km}$ wide and $150\,\mathrm{m}$ thick. The structure of the submesoscale eddies produced throughout the mesoscale eddy/topography interaction is consistent with the structure observed  *in the in situ* measurements. The fits of the horizontal and vertical profiles of PV anomaly, are close to Gaussian in both directions. The absence of sign reversal of the PV gradient is a clue to the stability of these small eddies (in the absence of external flow), thus their potentially long time life. Indeed, the ambient shear strain imposed by mesoscale eddy field is about $5\,10^{-6}\,\mathrm{s}^{-1}$. It corresponds to $\sim 10\% \times \zeta_{SCV}$ which is close to the quasi-geostrophic value necessary for vortices to elongate irreversibly.

Submesoscale eddies are either cyclonic or anticyclonic, opposite to the polarity of their parent mesoscale eddy. In this paragraph, we describe the lifecycle of two submesoscale eddies: a cyclone and an anticyclone (see Figure 8 and 9). This lifecycle is typical of the submesoscale eddies observed in the model simulations, generated in the BBL by interaction of the mesoscale eddies with the topography. Once formed, the submesoscale eddies rapidly merge with their neighbors (see Figure 9b, from

[revised manuscript text omitted]

Probability Density Function (PDF) of the normalized relative vorticity of the selected particles is shown in Figure 11b,c. The normalized relative vorticity associated with the submesoscale eddies dramatically decreases during the very first travel days. This corresponds to the merging events. Indeed, after merger, the submesoscale eddy formed is larger in size but is less intense (note that only potential vorticity is nearly conserved, in the limit of small viscosity). This relative vorticity loss occurs via

5  filamentation and spatial re-organization during the merging process; this re-organization is associated with relative vorticity to vortex stretching conversion, with an increase in potential energy (Ciani et al., 2016). The number of particles trapped in each eddy varies during the merging events. As the submesoscale eddies rotate around their parent mesoscale eddy, they are sheared and they lose particles at their periphery.

10  Finally, the impact of the submesoscale eddy production on the diffusion of particles is highlighted by comparing three experiments (i.e.: EXP1, EXP2 and EXP3). Particles are initially seeded at the southwest edge of the channel between 100 and 300 meter depths. Figure 12 shows the positions of particles with time.

In the EXP1, without BBL vorticity generation (Figure 12, 1st column), particles are advected mostly by mesoscale eddies. Particles remain mostly in the left hand side of the domain. In EXP2 and EXP3 (Figure 12, 2nd and 3rd columns respectively),

15  particles are advected by filaments over the sloping topography, subsequently by submesoscale eddies, and by mesoscale eddy as well. The dispersion of particles is then more efficient in EXP2 and EXP3 than in EXP1.

Following LaCasce (2008), we can estimate a dispersion coefficient as:

$$\kappa = \frac{1}{2}\frac{\mathrm{D}}{\mathrm{D}t} < D^2 > . \tag{14}$$

In Figure 13a, we show the time evolution of the relative dispersion $< D^2 >$ where $D$ is the distance between pairs of

20  particles and $< \bullet >$ is the ensemble average over all the pairs of particles. During the 25 first days of integration, the relative dispersion of particles is similar in the three experiments. Later, the relative dispersion increases more strongly in EXP2 and EXP3. Particles trapped within submesoscale eddies can reach the cyclones and anticyclones on the right side of the domain more easily and travel longer distances. The cape does not play a significant role in the particle dispersion due to the initial configuration of the mesoscale eddies. The estimations of the dispersion coefficient yield $\sim 1000\,\mathrm{m^2 s^{-1}}$ in both experiments

25  while in EXP1 it is about $700\,\mathrm{m^2 s^{-1}}$.

[Figure]

[Figure]

**Figure 13.** (a) Time evolution of the dispersion $< D^2 >$ computed for the three experiments: (green) EXP1, (blue) EXP2 and (orange) EXP3. (b) Probability Density Function of the relative vorticity normalized by the planetary rotation associated with particles regarding (green) EXP1, (blue) EXP2 and (orange) EXP3.

In Figure 13b,c,d, we show the PDF of the relative vorticity normalized by the planetary rotation associated with particles at day 200. In the three experiments PDFs are Gaussian shaped and have zero bias, meaning that the influence of cyclonic and anticyclonic motions is similar. However, in EXP2 and EXP3, PDFs flatten and large relative vorticity values are reached (i.e.: $|\zeta^z/f_0| > 0.5$), then the variances increase as a signature of the particle trapping in submesoscale eddies. Finally, particles can

5  also be trapped in the turbulent bottom boundary layer and remain in it for long time. This can be seen in Figure 12 in EXP2 and EXP3 at day 200. There, the bottom boundary layer vorticity is positive and large (i.e.: $|\zeta^z/f_0| > 1$.). That is why, PDF associated with EXP2 and EXP3 show that particles can have very large value of relative vorticity.

Finally, in Figure 14, we compare the number of particles transferred from the left to the right hand side of the channel. In EXP2 and EXP3, the amount of particles lying in the right hand side of the channel is $\sim 15\%$ of the total number of particles.

10  This amount of particles decreases to $\sim 5\%$ in EXP1. This result highlights the impact of submesoscale eddies on the path and spread of particles. Without subsurface submesoscale eddy generation, the fate of particles is driven by the mesoscale eddy field. In this case, particles are trapped in mesoscale eddies or recirculate at the periphery of mesoscale eddies. However, with the subsurface submesoscale eddy generation, particles can be trapped and advected by submesoscale eddies. Then, they can travel over large distance and long time. In EXP2, the amount of particles oscillates through time as a result of the recirculation

15  of particles due to the mesoscale cyclone lying in $x \in [300, 400]$ km. These oscillations do not appear in EXP3 which means that the cape does not allow particles to recirculate.

[Figure]

[Figure]

[Figure]

**Figure 14.** Time evolution of the percentage of particles lying in $x \in [400; 600]$km the three experiments: (green) EXP1, (blue) EXP2 and (orange) EXP3.

**4    Conclusions**

In situ observations have shown filaments and small eddies of Persian Gulf Water, formed from its outflow along the continental slope of the Gulf of Oman. At this time, a row of alternate signed mesoscale eddies was present in the gulf. We  _simulated_ numerically an idealized version of this configuration, corresponding to the interaction of a row of surface intensified mesoscale

5    eddies with two topographic slopes in a zonal channel. We used a high-resolution primitive equation model (a 3D hydrostatic model) and showed that the friction of these vortices on the slope (in a slanted bottom boundary layer) is the primary mechanism for the generation of opposite-signed relative vorticity on the slope. The vorticity filaments thus created then undergo shear instability and form a line of small eddies. These small eddies merge and grow in size. They also pair (cyclones with anticyclones) and propagate away from their region of formation. At the surface, these small eddies have a signature and can thus

10    be detected via satellite altimetry. This is due to their strong intensity and to the shallow depth at which they lie. Also, TRW breaking creates turbulence at depth. TRW propagate in the same (opposite) direction as the velocity field at the associated with the mesoscale cyclone (anticyclone). TRW destabilizes _via_ baroclinic instability as the velocity of the flow is opposite to its phase speed.

15    The trajectory of submesoscale vortices is often curved and their advection by the mesoscale eddy flow brings these small eddies back towards the topographic slope. There, these eddies generate smaller eddies in turn. This is manifested by a shallower slope of the enstrophy spectrum (more energetic small scales of vorticity). Finally, these small eddies can be destroyed by the shear of the mesoscale eddies (or be incorporated into them) or crash onto the topographic slope and be eroded there. Their lifetime is usually shorter than $3\,\mathrm{months}$ and the length of their pathway is at most $600\,\mathrm{km}$.

20    We have also shown that these small eddies play a non negligible role in the transport and dispersion of tracers (which can be salinity in the Gulf of Oman) by comparing two simulations with and without the BBL vorticity generation.  Due to
 submesoscale eddies, particles can travel over large distance more rapidly. Finally, the presence of a cape, such as the Cape of

[Figure]

Ra's al Hamra in the Gulf of Oman, does not impact significantly the diffusion of a passive tracer modeling the PGW.

This study also raises other worthwhile questions.

. In particular, we wish to use a non hydrostatic model to study  *in greater detail* the inter-
action between two SCVs or between one SCV and one mesoscale eddy.

5

Few filaments and eddies are observed in our simulation while no realistic surface forcings were implemented; in the ocean, filaments and small eddies can also originate from the turbulent interaction of eddies or from the instability of upwelling fronts. The vertical interaction between surface and subsurface SCVs is another  *Offshoot* of the present study.

*Acknowledgements.* We acknowledge support from ANR ASTRID Maturation project DYNED ATLAS and from UBO. We thank CNRS

[revised manuscript text omitted]

---

## Author Comment (AC1) · 13 Jul 2019

Dear referee,

Please find in attached the reponses to your comments and suggestions, and a new version of the MS.

Best regards,

Mathieu Morvan

Please also note the supplement to this comment:
https://www.ocean-sci-discuss.net/os-2019-3/os-2019-3-AC1-supplement.zip

---

## Author Response (AR1)

**Response to Referee #1**

**Major Comments**

*1. Section 3.3.1 needs some clarifying and expansion. First, as I understand the section and figure 5, you are simply arguing that the submesoscale eddies at 500 m depth are generated via baroclinic instability. This seems plausible, but should be put in the context of recent work on the topic (eg. Hetland 2017, and Wenegrat et. al 2018). Likewise, it was not clear to me whether the focus throughout on the mechanism being a topographic Rossby wave was meant to distinguish this in some way from the basic baroclinic instability mechanism over a slope (in which case this needs clarification), or whether it was just a particular way of introducing why baroclinic instability can happen over a slope (which I would argue is unnecessarily complicated and could just be replaced throughout by 'baroclinic instability').*
*It would also be good to dig a bit deeper in this section into related questions such as:*

- *Why is this mechanism not generating as active an eddy field at shallower depths in EXP1? A possible explanation might be the dependence of the instability on the Slope Burger number, such that the stronger stratification at shallower depths suppresses growth (Wenegrat et al. 2018).*
- *Is the instability trapped between the bottom and the pycnocline? Ie. what sets the vertical scale?*
- *What determines the separation of the eddies off the topography into coher- ent vorticies?*

Thank you for suggesting the paper written by Hetland (2017) and Wenegrat et al. (2018). We modified the section « 3.3.1 Unstable Topographic Rossby Waves » in « 3.3.1 Baroclinic instability at depth ». We re-wrote this section in order to make the process at play more clearer for the reader.

*2. The motivation of the study mentions both the Persian Gulf and Red Sea outflows, however the study is really only focused on the 200 m depth range (ie. the Persian Gulf water). For instance, both the detailed case studies and the particle tracking are focused only on the 100-300 m depth range. This choice may reflect the fact that it is only in this depth range where there are substantial differences between the experiments. However, the most active submesoscale eddy field is at 500 m depth (eg. figure 3).*
*As such, I would suggest that the particle tracking analysis should be repeated for 500 m depth. While there are likely not significant differences between experiments at this depth, the findings would have implications for the accuracy of lower-resolution models in capturing the spread of Red Sea water.*

In our configuration, the velocity field of mesoscale eddies reaches 300 m depth, typical of the Gulf of Oman. In the Gulf of Aden, the mesoscale eddies can reach 1000 m depth. So, the detachment of the bottom boundary layer should occur at depth as well and then, intense and long-lived submesoscale eddies should be created below 300 m depth and have an impact on the dispersion of particles (or the Red Sea outflow Water). That is why, we did not repeated the particle tracking at 500 m depth. However, we added couple of lines of discussion about this point in the conclusion (p. 19, l. 4).

*3. The differences in particle dispersion between the 3 experiments are being at- tributed to the submesoscale eddies in the interior. However, an alternate hypoth- esis would be that the differences arise due to boundary layer dynamics (absent in EXP1). Histograms are shown for vorticity field sampled by the particles (eg. figure 13 b, c, d), showing heavier tails in EXP2 and EXP3, which is interpreted as evidence of the role of submesoscale eddies. However, this same sort of pattern could also occur if the particles were randomly sampling the underlying flow field (which would have a heavier tailed vorticity distribution in EXP2 and EXP3). A bit more analysis of this section would make the argument for the role of submesoscale eddies more convincing. For example, one could look at the changes in the particle sampled vorticity distribution relative to the changes in the underlying distribution across the whole domain. You could also try comparing distributions between particles which make it to the right-hand side of the domain to those that don't.*

As you suggested, we compare now the distribution of vorticity associated with particles which reach the right-hand side of the domain to those which don't (See Fig. 14 and the text p. 16, l. 18).

**Minor Comments**

*1. You have high resolution in the vertical (100 σ levels), and moderately high- resolution in the horizontal, with moderately steep topographic slopes. Are hydrostatic pressure gradient errors a concern for this setup? It would be good to comment on this (eg. grid stiffness etc) in section 2.2.*

Yes, the hydrostatic pressure gradient errors are a concern for this setup. We added a comment about this (p.3 , l. 28), in particular, regarding our sensitivity experiments.

*2. You should comment a bit more on the choice to model the dispersion of dense water as a passive tracer. As I understand the setup, really what you are intending to say is that the passive particles are meant to act as a proxy for high-salinity water. I assume that this choice was made because introducing a salinity gradient in the initial condition would be problematic with the re-entrant domain.*

We did not add a dense water directly in our initialization for several reasons. Firstly, as mentioned in the MS now (p. 6, l. 10), there is no submesoscale velocity structure clearly associated with the fragment of Persian Gulf outflow Water (observation made from the VM-ADCP). It indicates the PGW can be modeled as a passive tracer. Secondly, as pointed out, the periodic boundary condition in the x-direction would have been problematic. Thirdly, we are not interested in the dynamics of the outflow itself. We added some comments (p.5 , l. 25).

*3. Given that you only have 3 runs, I would suggest renaming them with more in- formative names, which is very helpful to the reader. For example, you could choose to name them NO-BBL, BBL, and BBL-CAPE, or any other variant that immediately conveys the setup.*

We re-named the experiments in the text, captions and figures as you suggested.

*4. In section 3.5 you introduce two different definitions of the diffusivity (equations 13 and 14). Please clarify why these two definitions are given, why they don't agree, and if possible clean this section up a bit by using only one.*

As mentioned in the MS, the first definition allows to estimate a diffusivity coefficient for short times. This is the case of our ballistic regime. Then, thanks to the second definition, we can estimate a diffusivity coefficient over a long time period.

*5. The final paragraph of the manuscript feels out of place, and not well supported. For example 'the vertical motions then are of importance to the uplift of nutrients in the ocean and then onset of algae blooms' is extremely speculative when considering an instability at 200 m depth. As this paragraph really is just laying out a variety of future work the authors plan to carry out, it is not entirely relevant to the bulk of the manuscript, and I'd suggest removing it.*

The final paragraph has been rewritten thanks to the referes suggestions.

6. In some of the figures the subplots lack labels/scales on the axes. For instance, figure 8 shows an eddy in plan view, without axes labels. The moving focus region between subplots would make it hard to label with absolute position, however you could at least add some scale to the x-y axes (ie are the subplots showing a 10km x 10 km region? 100 km x 100 km?).

This has been modified.

*7. Figure 10: Describe the meaning of the dashed lines in the caption.*

This has been added.

*8. The wording in the abstract connecting the findings here to the Persian Gulf Water and Red Sea water is a bit too strong. I would suggest rewording to: ...and their potential impact on the spread of Persian Gulf Water..' and 'This shows the potentially important role of submesoscale eddies...'.*

This has been modified in the MS.

*9. Spelling: 'without' near line 15 in the abstract.*

We modified this.

*10. I assume that the black contours in Figure 2 (b) and (c) are density, however this is not indicated in the caption.*

This was added in the caption.

**RESPONSE TO REFEREE #2**

1. *Section 2.1 and 3.1: Did the Physindien 2011 campaign show evidence of submesoscale flows? While there is some evidence in the GO13 (figure 2c) section to support the filaments of Persian Gulf Water flow in T and S, but there is no velocity information presented from the corresponding ADCP sections. I would like the authors to present some velocity information from the observational campaign in section*

2. *What scales do you see (after suitable averaging in the ADCP section? The accuracy of horizontal velocity components is mentioned, but no data is shown from the ADC section so this is superfluous information.*

3. *Do you see any evidence of submesoscale vortices in the velocity structure from the observations? At what depths? How high were the 2-d vorticity that you could observe in the ADCP sections? At what depth?*

[Figure]

Figure 1. (a) Map of Absolute Dynamic Topography anomalies in the Gulf of Oman averaged over the duration of the cruise processed by CLS-Argos on a **⅛** degree Mercator grid. The solid grey lines indicate the locations of the Seasoar sections. Vertical sections of (From top to bottom) temperature, salinity, zonal and meridional velocity components of the GO16 (left column) and GO13 (right column) sections.

Figure 1 shows the zonal et meridional velocity components measured by the VM-ADCP. We did not present any velocity informations as there is no submesoscale flow clearly associated with the fragments of Persian Gulf outflow Water. The submesoscale velocity field of the ADCP is noisy, so we had to smooth coarsely the velocity fields so that we did not see any evidence of submesoscale flow at the depth of the Persian Gulf outflow Water. We added a comment in the MS (p. 6, l.10).

*4. What do the Observed KE spectra show at various depths for the sections shown in Figure 2. In the ranges that overlap âAˇTˇ how do the spectra from the model and the observations compare (in terms of slopes, etc.?)?*

The comparison of the KE spectra slopes between the model and *in situ* measurements is not reliable since we do not have many points in the VM-ADCP sections and the measurements are very noisy. But, if we had better measurements, this would be a great idea and useful information.

*5. Page2, Lines 27-28: The simulations you are doing are very idealized and you are using the observations as an inspiration, so saying that the simulations "specifically designed to resemble the local geography of the Gulf of Aden and the Gulf of Oman" is incorrect. The geometry is very idealized, and the Gulf is variable in width unlike your simulations and has many other geographical details. You need to modify this sentence to reflect this.*

**We agree and we modified this in the MS.**

*6. Page 3, Line 17: The accuracy of ADCP u,v is mentioned to be 0.5cm/s. With what averaging? No ADCP section is shown, and no averaging information is mentioned, so this information is not useful to your reader.*

We added in the MS (p. 3, l. 15) some informations about the VM-ADCP.

*7. Figure 2a). Change the color for the GO13 section, it is very hard to see the line corresponding to GO13 in this panel.*

This has been changed and we are sorry about this.

*8. Figure 3. We can see intense vorticity variations in this figure at sub-mesoscales. Does the density stratification vary on these scales? You should present either density or N^2 from EXP1,2 and 3 at these depths as well âAˇTˇ are these vortices seen in the density structure as well?*

We now show in Fig. 7, the vertical section of PV anomaly of a cyclonic and anticyclonic submesoscale eddies. We added the contours of the density anomaly associated with both submesoscale eddies.

*9. Page 8Line 15, and Page 9 Line 1-5: We know from your initial conditions and from the observations that the flow is stratified. Why does the "homogeneous" fluid TRW match the observed phase speeds? What is the phase speed if stratification was taken into account?*

We added a sentence mentioning this in the text (see section 3.1.1).

10. Section 3.5 and 4: Particularly for the last part of section 4, the presentation of the text needs to be improved, and I have offered several suggestions in the attached annotated version of the ms.

Thank you very much, your suggestions are very helpful and we took them into account.

*Please also note the supplement to this comment: https://www.ocean-sci-discuss.net/os-2019-3/os-2019-3-RC2-supplement.pdf*

We modified the MS by taking this into account.

**Response to Short Comment #1**

*The major shortcoming is a poor comparing the simulation results with natural measurements. How to extent the considered mechanisms can characterize the spreading the Persian Gulf Water and Red Sea Water?*

In our simulations, we did not implement any outflow waters such as the Persian Gulf or Red Sea outflow waters. So, it is difficult to extent directly the comparison between in situ measurements and results from the numerical experiments. However, we added in the MS some details about the results from the VM-ADCP measurements; in particular, no submesoscale flow clearly associated with the fragments of Persian Gulf Water was observed.

*As well as, authors do not compare their results with that from the other numerical simulation studies.*

We cited the paper wrote by Vic et al. (2015) (p. 8, l. 4 in the MS) who have shown that submesoscale eddies were produced at subsurface due to the drag of mesoscale eddies over the sloping topography.

*The model configuration description is very brief. Please specify the used parameterizations of subgrid-scale, open boundary conditions, momentum fluxes, heat and salt fluxes on the upper boundary of the channel.*

We added sentences in the MS starting (p. 3, l. 23) about the model configuration.

*Please compare the observed temperature and salinity profiles with these from the numerical simulations. What differences between them exist?*

We are not able to do this since we did not implement any outflow waters.

*The generation period of intensive submesoscale variability is about 200 days. During this period the background conditions: the spatial structure of mesoscale eddies and temperature and salinity can change. Please give proof the stability of the mesoscale eddy row during 200 days.*

In the MS, we did not say that mesoscale eddies are long lived. Indeed, they may not be, since they interact with the sloping topography. However, their intensity as well as their size are similar from the beginning till the end of the simulations.

**SHORT COMMENTS**

*1. Line 15. Please change, Figure 5a on Figure 5d.*

We did not change this, since now, Figure 5a appears before Figures 5d in the text.

*2. Please give the definition of the SCV.*

We gave the definition (p. 12, l. 10).

[revised manuscript text omitted]

---

## Author Response (AR2)

**RESPONSE TO ANONYMOUS REFEREE #1**

**(REPORT #2)**

*The authors have provided a substantive revision, and satisfactorily addressed my prior comments. I have some very small (trivial) suggestions that the authors might consider (listed below), but feel the manuscript is very nice, and acceptable regardless of whether they choose to make these changes or not.*

*Minor Comments:*

*1) Page 2 line 31: I still find the wording somewhat confusing, in that you refer to 'dense water' which you will model as a 'passive tracer', which of course it is not. I think you could clean this up (here and other places) a bit more by emphasizing that you will show that submesoscale eddies enhance passive particle dispersion in your model, which then has implications for understanding other forms of dispersion of particular watermasses, etc. (The subtlety here being that you are not showing anything directly about dense water dispersion).*

We modified the sentence page 2, line 32 which is now: « We further quantify how these submesoscale eddies are instrumental in the spreading of passive particle, then the implications regarding the dispersion of dense waters such as PGW and RSW. »

And page 14, line 6 : « We examine the capability of the submesoscale vortices to carry a tracer, such as the anomalously high salt content associated with the Persian Gulf Water and the Red Sea Water. » is now « PGW and RSW are dense waters due to their anomalously high salt content. As revealed by in situ observations, the filaments of PGW are pulled off the coast by mesoscale eddies and break into submesoscale eddies containing PGW. In this section, we examine the capability of the submesoscale vortices to carry a passive tracer. »

*2) Following up on my prior review minor comment #4, I don't think you've fully clarified in the text why you have 3 different definitions of diffusivity. In the reviewer response you mention what you use them for (short times, ballistic phase, long times) but don't give the reader (who may not be familiar with the particle dispersion literature) any sense of why different definitions are needed in each case. For instance, only the form given in (14) is familiar to me. On this note, I would also try to be a bit more clear that you transition in this one section from discussing dispersion by the 2 particular SCVs to dispersion of particles seeded in the SW part of the domain.*

In the MS we explicitly wrote page 14, line 11 « For short times, a dispersion coefficient ... » about the first definition, and page 16, line 5 « Following LaCasce (2008), we can estimate an equivalent diffusivity coefficient through time as : ... ».

*3) On page 19, paragraph starting on line 8: This is improved from the prior version, but it still feels a bit out of place, and not as strongly supported as the rest of the manuscript. For example, the claim that interaction between a SCV and a mesoscale eddy in a non-hydrostatic model 'will be important for onset of algae blooms' seems very speculative and non-obvious. If anything it would seem that interaction between surface mixed-layer eddies and SCVs (as you discuss at the end of the paragraph) might be a better candidate for generating enhanced nutrient fluxes. I would suggest taking another pass through these last two paragraphs and trying to improve the wording, and make*

*them a little less speculative, as the study you present here is interesting without these broader speculations.*

We modify the end of the conclusion page 19,line 9 by taking into account this point.

**Response to Anonymous Referee #2**

**(Report #1)**

*The description of the Physindien2011 campaign has details of the 38kHZ ADCP data, which is not presented or used in the paper. Therefore I would recommend that you take out the following sentences from section 2.1, page3, lines 14 to 16: "The horizontal velocity...5\times10^-3 m s^-1." This is just superfluous information.*

We decided to keep this information since it was required in another review.

[revised manuscript text omitted]

*In situ* measurements and detection of SCVs remain sparse. Nevertheless we are now able, thanks to the satellites data, to localize the mesoscale eddies and their distances from the coast. The conjoint use of these maps, and our estimates of the SCVs positions relative to mesoscale eddies and topography, must lead us to perform experiments at sea in both gulfs to capture the
15  characteristics of the SCVs.

*Acknowledgements.* We acknowledge support from ANR ASTRID Maturation project DYNED ATLAS and from UBO. We thank CNRS and RFBR for support under PRC project 1069 (in French classification) and 16-55-150001 (in Russian classification). Mikhail Sokolovskiy was supported also by the Ministry of Education and Science of the Russian Federation (Project No.14.W.03.31.0006)

**References**

[revised manuscript text omitted]